# Biocontrol efficiency and mechanism of novel *Streptomyces luomodiensis* SCA4-21 against banana *Fusarium* wilt

Qiao Liu,[1] Liangping Zou,[1] Yufeng Chen,[1] Junting Feng,[1] Yongzan Wei,[1] Miaoyi Zhang,[1] Kai Li,[1] Yankun Zhao,[1] Dengbo Zhou,[1] Wei Wang,[1] Dengfeng Qi,[1] Jianghui Xie[1]

**ABSTRACT**  The soil-borne fungus *Fusarium oxysporum* f. sp. cubense tropical race 4 (Foc TR4) causes banana *Fusarium* wilt, a serious threat to global banana production. Biocontrol represents a promising alternative to conventional chemical control for managing this disease. Our previous studies identified *Streptomyces luomodiensis* SCA4-21 as a novel species with antifungal properties. Here, we show that the extract from strain SCA4-21 significantly inhibits Foc TR4 mycelial growth and spore germination, severely disrupting hyphal and spore ultrastructure. The extract also exhibited broad-spectrum activity against eight other phytopathogenic fungi. We identified 32 volatile organic compounds produced by strain SCA4-21, including five with known antifungal properties. In pot experiments, strain SCA4-21 inoculation not only significantly suppressed Foc TR4 infection in banana seedlings, with a biocontrol efficacy of 59.3%, but also markedly promoted plant growth. The inoculation significantly enriched beneficial bacterial genera (*Streptomyces*, *Bacillus*, *Sphingomonas*, *Massilia*, and *Pseudomonas*) and fungal genera (*Mortierella*, *Gibellulopsis*, and *Xenomyrothecium*), while reducing the pathogenic bacterium *Pantoea* and lowering *Fusarium* abundance to levels statistically indistinguishable from the control. Microbial function prediction based on 16S rRNA gene sequencing data revealed enhanced metabolic pathways, including carbohydrate metabolism, amino acid metabolism, and metabolism of terpenoids and polyketides. We propose that strain SCA4-21 combats banana *Fusarium* wilt through a synergistic mechanism involving antifungal compound production and recruitment of beneficial microbiota. These findings highlight strain SCA4-21 as a promising biocontrol agent for sustainable banana production.

**IMPORTANCE**  Banana (*Musa* spp.) is one of the most popular fruit crops and the fourth largest food crop in developing countries within tropical and subtropical regions. However, the emergence and rapid spread of strain *Fusarium oxysporum* f. sp. cubense tropical race 4 (Foc TR4) seriously hinder the development of the banana industry. Currently, there is no effective control measure available. Biological control holds potential due to its safety and effectiveness. Here, we found that the extract of *Streptomyces luomodiensis* SCA4-21 exhibited significant inhibitory effects on the hyphal growth and spore germination of Foc TR4, as well as severe destructive effects on its cell morphology and ultrastructure, and broad-spectrum antifungal activity. We also discovered that the inoculation of strain SCA4-21 significantly inhibited the infection of Foc TR4 in banana seedling corms, reduced the incidence index of banana *Fusarium* wilt, promoted the growth of banana seedlings, and enhanced beneficial microbes and metabolic pathways, suggesting that strain SCA4-21 is a promising biocontrol agent.

**KEYWORDS**  *Streptomyces luomodiensis*, biocontrol, banana *Fusarium* wilt, soil microbiome, extract, antifungal activity

Address correspondence to Wei Wang, wangweisys@ahau.edu.cn, Dengfeng Qi, qidengfeng@itbb.org.cn, or Jianghui Xie, xiejianghui@itbb.org.cn.

Qiao Liu and Liangping Zou contributed equally to this article. Author order was determined based on their primary contributions.

The authors declare no conflict of interest.

Banana is a major source of starch and a vital economic crop in tropical and subtropical developing countries (1). However, banana production is severely threatened by fungal diseases, particularly *Fusarium* wilt. This disease, caused by *Fusarium oxysporum* f. sp. cubense tropical race 4 (Foc TR4), is among the most devastating soil-borne pathogens globally. The global expansion of Foc TR4 is accelerating, with its total economic impact estimated at up to $10 billion (2). The devastation caused by Foc TR4 stems from both its ability to infect almost all banana cultivars and its long-term survival capacity (3). Even in the absence of living host tissues, Foc TR4 can survive in soil as chlamydospores for over 20 years. Once the environmental conditions become favorable, the pathogen invades the root system of banana plants and spreads rapidly. Its proliferating mycelium and secretions obstruct the vascular system, inducing water stress that leads to wilting and eventual mortality of the plant (4).

The utilization of biocontrol agents for managing banana *Fusarium* wilt is regarded as a viable alternative to conventional chemical control owing to its efficacy and eco-friendliness (5). *Streptomyces* spp. are ideal candidates for biocontrol agents against soil-borne diseases. This is largely owing to their remarkable resilience, allowing them to survive for long periods under extreme conditions, such as low nutrition and water availability, in the form of spores (6). In particular, rhizosphere-associated *Streptomyces* spp. possess the added advantage of being able to actively colonize plant root systems, providing a critical line of defense at the infection site (7–9). They have been studied extensively as potential biocontrol agents against soil-borne phytopathogens such as *Ralstonia solanacearum*, *F. oxysporum* f. sp. *cubense Race 1*, *F. oxysporum* f. sp. *cucumerinum,* and *Rhizoctonia solani* (10–12). They protect crops from diseases by synthesizing diverse bioactive secondary metabolites, competing with pathogens for nutrients by producing substances like siderophores, or producing a large number of extracellular enzymes related to fungal cell wall degradation (12–15).

The exploration of rhizosphere-derived *Streptomyces* strains represents a promising strategy for biologically controlling *Fusarium* wilt of banana caused by Foc TR4. Our research has systematically advanced from initial phenotypic screening to the mechanistic dissection of antagonistic strains from extreme environments. It began with the isolation of *Streptomyces* sp. CB-75 from a conventional banana rhizosphere—a strain that demonstrated excellent efficacy in suppressing disease and promoting plant growth (16). To further pursue strains with novel biosynthetic potential, we then targeted the underexplored dry-hot valley ecoregion. This approach led to the discovery of two novel rhizospheric species: *Streptomyces huiliensis* SCA2-4, whose genome harbors 51 biosynthetic gene clusters and whose extract strongly disrupts Foc TR4 cellular integrity (17), and *Streptomyces sichuanensis* SCA3-4, which employs a multifaceted mode of action, including siderophore-mediated competition and induction of apoptotic cell death in the pathogen (18).

We subsequently discovered the novel rhizospheric *Streptomyces luomodiensis* SCA4-21 from this same environment, a strain that exhibits potent broad-spectrum *in vitro* antifungal activity, including against Foc TR4 (19). However, its overall biocontrol potential *in vivo* and the mechanisms underlying its antifungal effect remain unknown. Therefore, the objective of this study was to comprehensively evaluate the biocontrol potential of *S. luomodiensis* SCA4-21 and elucidate its antifungal mechanisms. To achieve this, we first characterized its antifungal compounds and *in vitro* efficacy by determining the broad-spectrum antifungal activity of its extract against a panel of phytopathogens; characterizing the extract's inhibitory effects on the cellular and ultrastructural morphology of Foc TR4; and identifying the chemical composition of its antifungal metabolites, particularly volatile organic compounds.

Moving beyond *in vitro* analysis, we then sought to evaluate the strain's performance as a sustainable biocontrol agent in a plant-soil system. Central to this approach was the application of the live strain SCA4-21. We reasoned that as a rhizosphere-competent *Streptomyces*, its introduction into the soil would enable a sustained, on-site interaction with the pathogen and plant roots. This is anticipated to facilitate the

continuous and *in situ* production of its antifungal metabolites directly within the infection zone, thereby offering a more durable and context-dependent protective effect than a single application of purified compounds. Furthermore, we hypothesized that the live *Streptomyces* could exert indirect biocontrol by modulating the soil microbiome. Consequently, we assessed the biocontrol efficacy and growth-promoting effects of live strain SCA4-21 on banana seedlings and analyzed its impact on the native soil microbial community to explore its potential to shape a disease-suppressive microbiome.

## RESULTS

### Extraction and purification of strain SCA4-21 extract

The ethanol extract of strain SCA4-21 fermentation broth was fractionated using macroporous resin chromatography with stepwise gradients of methanol (50%, 60%, 70%, and 100%). The antifungal activity of each fraction against Foc TR4 was evaluated based on the mycelial inhibition rate, with 10% dimethyl sulfoxide (DMSO) as the reference control (Fig. 1A).

The results demonstrated a clear positive correlation between the methanol concentration used for elution and the antifungal efficacy. The fraction eluted with 100% methanol exhibited the strongest inhibition at $(71.57 \pm 0.08)\%$, followed by those eluted with 70%, 60%, and 50% methanol, with inhibition rates of $(31.53 \pm 0.15)\%$, $(17.67 \pm 0.25)\%$, and $(9.39 \pm 0.17)\%$, respectively. For comparison, the positive control, 33.3 µg/mL tebuconazole (in 10% DMSO), showed an inhibition rate of $(83.70 \pm 0.01)\%$. Consequently, the 100% methanol fraction, which displayed the most potent activity, was selected for subsequent investigation.

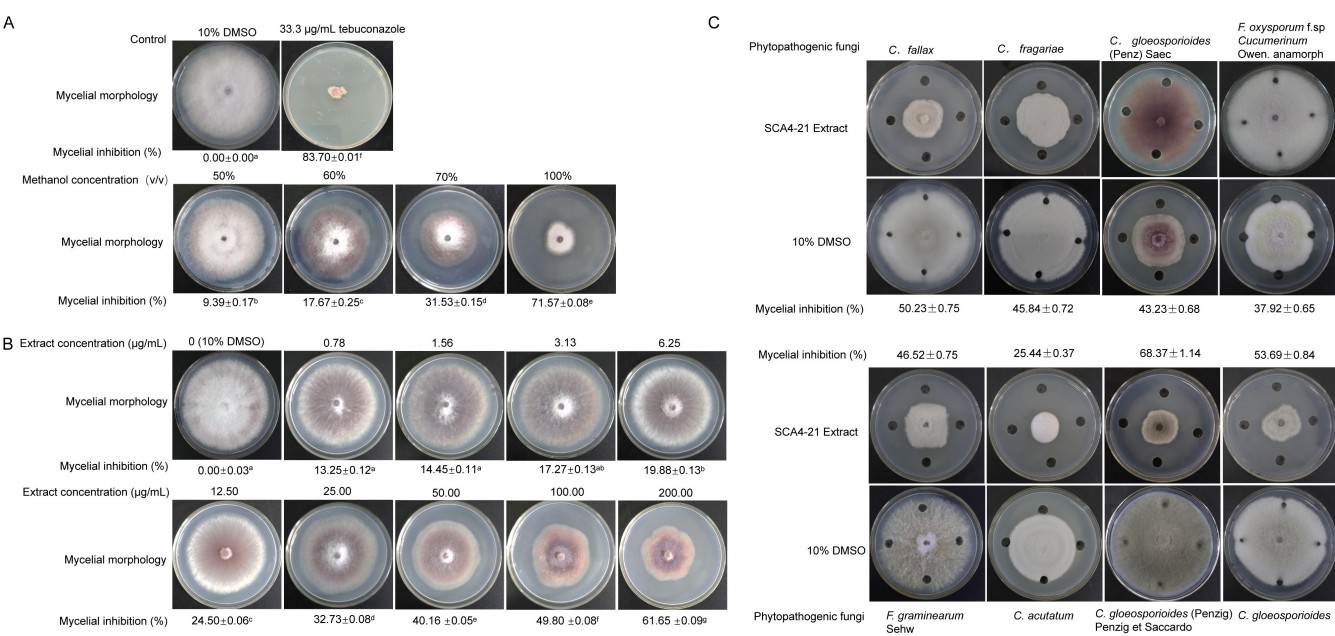

FIG 1 The 100% methanol fraction of *S. luomodiensis* SCA4-21 exhibits potent and broad-spectrum antifungal activity. (A) Fractionation of extract with methanol gradients (50%, 60%, 70%, and 100%) and their inhibition of *Fusarium oxysporum* f. sp. *cubense* tropical race 4 mycelial growth. The active 100% methanol extract was used for subsequent experiments. (B) Dose-dependent inhibition of Foc TR4 by the extract. (C) Broad-spectrum activity of the extract (100 µg/mL in potato dextrose agar) against eight other phytopathogens. For all panels, activity is shown as mycelial growth inhibition (%). Controls: 10% DMSO (negative) and 33.3 µg/mL tebuconazole (positive). Data are means ± SD ($n = 3$); different letters indicate significant differences ($P < 0.05$, Duncan's test).

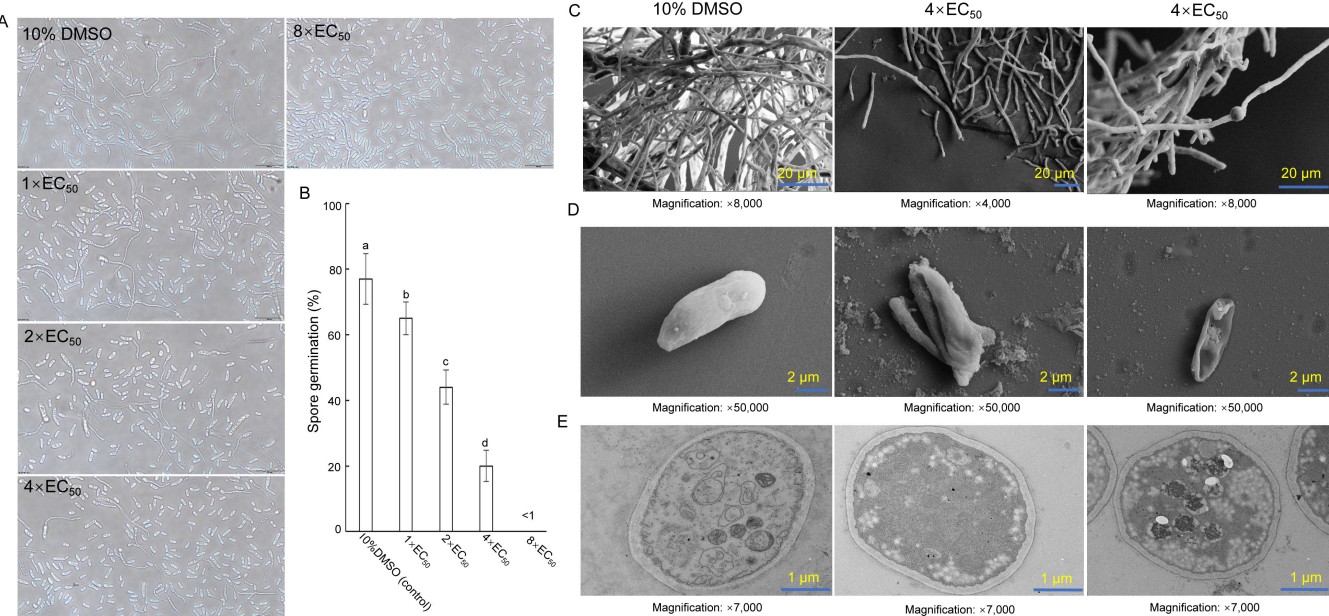

**FIG 2** *S. luomodiensis* SCA4-21 extract inhibits germination and disrupts the cellular integrity of Foc TR4. (A) Representative inverted microscopy images showing the dose-dependent inhibition of Foc TR4 spore germination by SCA4-21 extract (1× to 8× $EC_{50}$) after 24 h of treatment at 28°C. (B) Quantitative germination rates. A total of 100 spores were counted per biological replicate (100 spores × 3 microscopic fields; $n$ = 3 replicates). Different letters indicate significant differences ($P < 0.05$, Duncan's test). (C) Scanning electron microscopy (SEM) images of hyphae treated with SCA4-21 extract (4× $EC_{50}$) for 72 h, showing swelling and fusion. (D) SEM images of spores treated with SCA4-21 extract (4× $EC_{50}$) for 24 h, revealing a concave and disrupted morphology. (E) Transmission electron microscopy images of hyphae treated with SCA4-21 extract (4× $EC_{50}$) for 72 h, revealing severe ultrastructural damage. Scale bars are included in images. The control groups were treated with 10% DMSO.

## Measuring the $EC_{50}$ of strain SCA4-21 extract on the mycelial growth of Foc TR4

The half-maximal effective concentration ($EC_{50}$) of strain SCA4-21 extract on mycelial growth of Foc TR4 was determined. Mycelial inhibition showed a dose-dependent effect. After exposing Foc TR4 to the extract at 28°C for 7 days, significant growth inhibition of the pathogen was observed in all concentration groups (0.78, 1.56, 3.13, 6.25, 12.5, 25, 50, 100, and 200 mg $L^{-1}$). The inhibition of mycelial growth was as follows: 13.25 ± 0.12, 14.45 ± 0.11, 17.27 ± 0.13, 19.88 ± 0.13, 24.50 ± 0.06, 32.73 ± 0.08, 40.16 ± 0.05, 49.80 ± 0.08, and 61.65 ± 0.09, respectively. $EC_{50}$ value is calculated as 37.88 ± 1.58 µg/mL using toxicity regression equation (Fig. 1B).

## Strain SCA4-21 extract exhibits broad-spectrum antifungal activity

Strain SCA4-21 extract showed effective antifungal activity against the tested strains of fungi. Compared to the control treated with 10% DMSO, the fungal hyphae treated with the extract displayed significant inhibition. The inhibition rates for *Colletotrichum gloeosporioides* (Penzig) Penzig et Saccardo, *C. gloeosporioides, Curvularia fallax, Fusarium graminearum* Sehw, *C. fragariae, C. gloeosporioides* (Penz) Saec, *F. oxysporum* f. sp. *cucumerinum* Owen. anamorph, and *Colletotrichum acutatum* were 68.37%, 53.69%, 50.23%, 46.52%, 45.84%, 43.23%, 37.92, and 25.44, respectively (Fig. 1C).

## Strain SCA4-21 extract significantly inhibits spore germination of Foc TR4

The germination of spores was significantly inhibited after being treated with extract from strain SCA4-21 (Fig. 2A). The inhibitory efficiency showed a positive correlation with the concentration of the extract used in the treatment. Compared to the control group, the spore germination rates of Foc TR4 were 65%, 44%, and 20% after treatment with

$1\times$ EC$_{50}$, $2\times$ EC$_{50}$, and $4\times$ EC$_{50}$, respectively. The inhibition efficiencies were calculated as follows: 15.6%, 42.9%, and 74.0%. Furthermore, almost complete inhibition of spore germination was observed when an extract concentration equivalent to $8\times$ EC$_{50}$ was used (Fig. 2B).

## Strain SCA4-21 extract severely destroys the morphology and ultrastructure of Foc TR4

Scanning electron microscopy (SEM) observation revealed that the control mycelium of Foc TR4 treated with 10% DMSO exhibited a smooth and intact morphology, while mycelium of Foc TR4 treated with the $4\times$ EC$_{50}$ extract displayed notable alterations such as swelling and fusion (Fig. 2C). Simultaneously, the treated spores also exhibited a disrupted and concave morphology, whereas the control spores displayed a regular and planar appearance (Fig. 2D).

Transmission electron microscopy analysis showed that the ultrastructure of Foc TR4 hyphae with the $4\times$ EC$_{50}$ extract was also seriously damaged. Their organelles, such as mitochondria and vacuoles, disintegrated. The cytoplasm shrinks, the cell membrane becomes indistinct, and the nucleus disappears. In contrast, the cell membrane and wall of the control group were clear and complete. Cell vacuoles are filled, and mitochondrial cristae are clearly distinguishable (Fig. 2E).

## Identification of the volatile organic compounds of strain SCA4-21

Gas chromatography coupled with mass spectrometry (GC-MS) analysis was conducted to identify the volatile organic compounds (VOCs) of strain SCA4-21 that are likely responsible for its antifungal activities. By comparing the mass spectra of treatments with those of controls, 32 secondary metabolites of *S. luomodiensis* SCA4-21 were determined, including (2R,3R,4S)-2,3,4-trimethyloxetane, methyl butanoate, fluoroethene, methyl 3-methylbutanoate, methyl (E)-2-methylbut-2-enoate, methyl 3-methylpentanoate, methyl 4-methylpentanoate, methyl hexanoate, 3,3-dimethylpropanethioic S-acid, 6-Methyl-2-heptanone, 5-methyl-2-heptanone, methyl (2E)-2-hexenoate, 1-(2-furyl)-2-hydroxyethan-1-one, 3-octanone, methyl 5-methylhexanoate, and methyl heptanoate. (Fig. 3; Fig. S1A through F; Table S1).

## Strain SCA4-21 significantly reduces the incidence of banana *Fusarium* wilt disease

The pot experiment revealed that strain SCA4-21 significantly reduced the incidence of banana wilt disease caused by Foc TR4 (Fig. 4A and B). Specifically, no evident chlorotic leaves were observed in the negative control group (T1) where banana plantlets were treated with sterilized soybean liquid medium (SLM) only. While in the positive control group (T2), the leaves of banana plantlets treated with sterilized SLM and GFP-Foc TR4 exhibited serious chlorotic symptoms, resulting in a disease index of 57.5%. In contrast, only a few old leaves of banana seedlings treated with GFP-Foc TR4 + SCA4-21(T3) showed yellow symptoms, leading to a significantly lower disease index of 23.4% compared to positive controls (Fig. 4A). The biocontrol efficiency of strain SCA4-21 against banana *Fusarium* wilt was 59.3% (Fig. 4B).

Similarly, the corm sections of the T1, T2, and T3 groups displayed white, dark brown, and light brown colors, respectively. Laser confocal microscope analysis further revealed that only a small amount of mycelium was observed in the outer cells of the T3 bulbs, unlike the corms in the T2 group, where vascular cells were filled with Foc TR4 hyphae, while the corms in the T1 group were not infected by Foc TR4 (Fig. 4A).

## Strain SCA4-21 significantly promotes the growth of banana seedlings

The pot experiment also indicated that strain SCA4-21 significantly promoted the growth of banana seedlings (Fig. 4C through H). The T3 group, treated with SCA4-21 + Foc TR4, exhibited significant increases in plant height (62.9%) (Fig. 4C), stem diameter (25.2%)

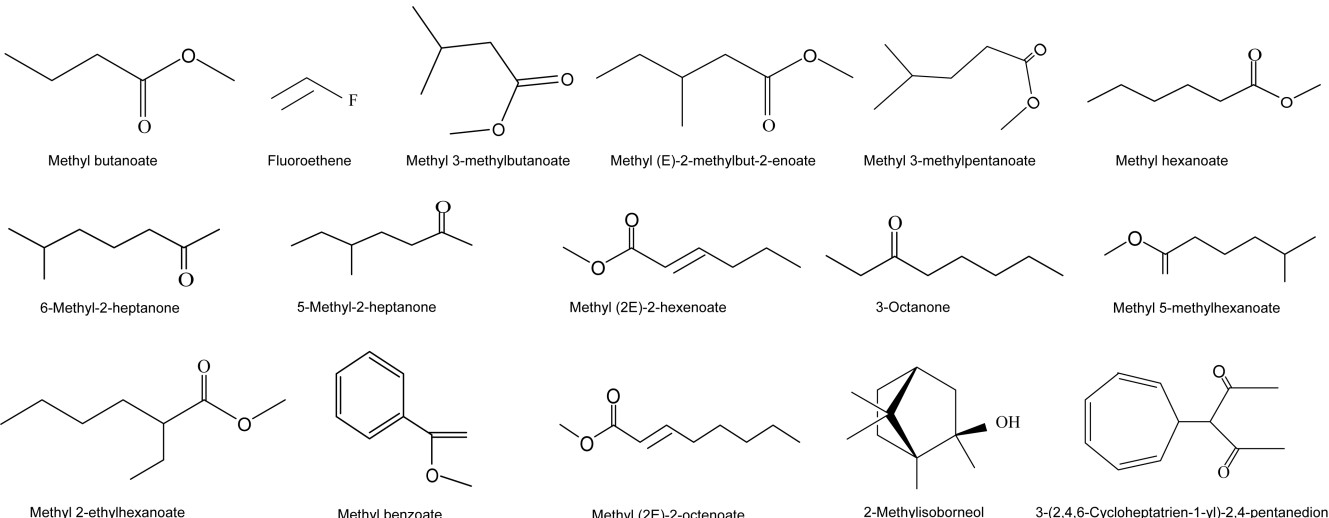

**FIG 3** *S. luomodiensis* SCA4-21 produces a diverse suite of antifungal volatile organic compounds. VOCs were collected from the headspace of a 7-day strain SCA4-21 culture in soybean liquid medium (SLM) using solid-phase microextraction. Comparative analysis with the sterile SLM control led to the identification of 32 secondary metabolites. Shown here are 16 representative structures from this set, which include key compounds such as various methyl esters and ketones postulated to contribute to the antifungal activity.

(Fig. 4D), chlorophyll content (24.3%) (Fig. 4E), leaf area (43.9%) (Fig. 4F), vane thickness (14.9%) (Fig. 4G), fresh weight (115.5%), and dry weight (59.3%) of banana seedlings 60 days after planting when compared to the T2 group treated with SLM + Foc TR4 (Fig. 4H).

Compared to the T1 group treated with SLM, the T3 group also showed significant increases in plant height (23.3%), fresh weight (25.6%), and dry weight (9.6%). Additionally, stem diameter, chlorophyll content, leaf area, and leaf thickness increased by 5.9%, 9.1%, 8.9%, and 2.8%, respectively; however, there was no statistically significant difference between these four indicators in the two groups.

## Strain SCA4-21 shapes the structure of the banana rhizosphere bacterial community

A total of 720,187 pairs of raw reads were obtained from nine soil samples by sequencing the V3 + V4 regions of bacterial 16S rRNA. Subsequently, after filtering, cutting, and splicing, a total of 718,594 pairs of clean reads were obtained (Table 1). After the chimera was removed, the obtained 482,135 non-chimeric clean reads were clustered into 11,210 distinct bacterial amplicon sequence variants (ASVs). Among them, 1,732, 1,899, and 1,862 ASVs were derived from T1 group soil samples treated with sterilized SLM, while 1,946, 1,863, and 1,851 ASVs were derived from T2 group soil samples treated with sterilized SLM and GFP-Foc TR4. Similarly, 1,602, 1,852, and 1,647 ASVs were derived from T3 group soil samples treated with Foc TR4 + SCA4-21 (Fig. 5A). Venn analysis revealed that T1, T2, and T3 groups had 3,201, 3,408, and 3,145 unique bacterial ASVs, respectively (Fig. 5B).

Analysis of bacterial alpha diversity showed that the Shannon and Simpson diversity indices in T1 group soils were higher than those in T2 and T3 group soils (Fig. 5C and D). In contrast, T2 group soils showed higher richness indices (ACE and Chao1), followed by the T1 and T3 group soils (Fig. 5E and F). However, there were no statistically significant differences observed in the bacterial diversity indices Shannon and Simpson, as well as the richness indices ACE and Chao1, among the three treatments (Fig. 5C through F; Table S2).

Principal component analysis (PCA) revealed that the primary source of variation in bacterial communities was the difference between treatments (Fig. 5G). The first two principal components (PC1 and PC2) together explained 84.46% of the total variance (66.89% and 17.57%, respectively), adequately capturing the community structure. While

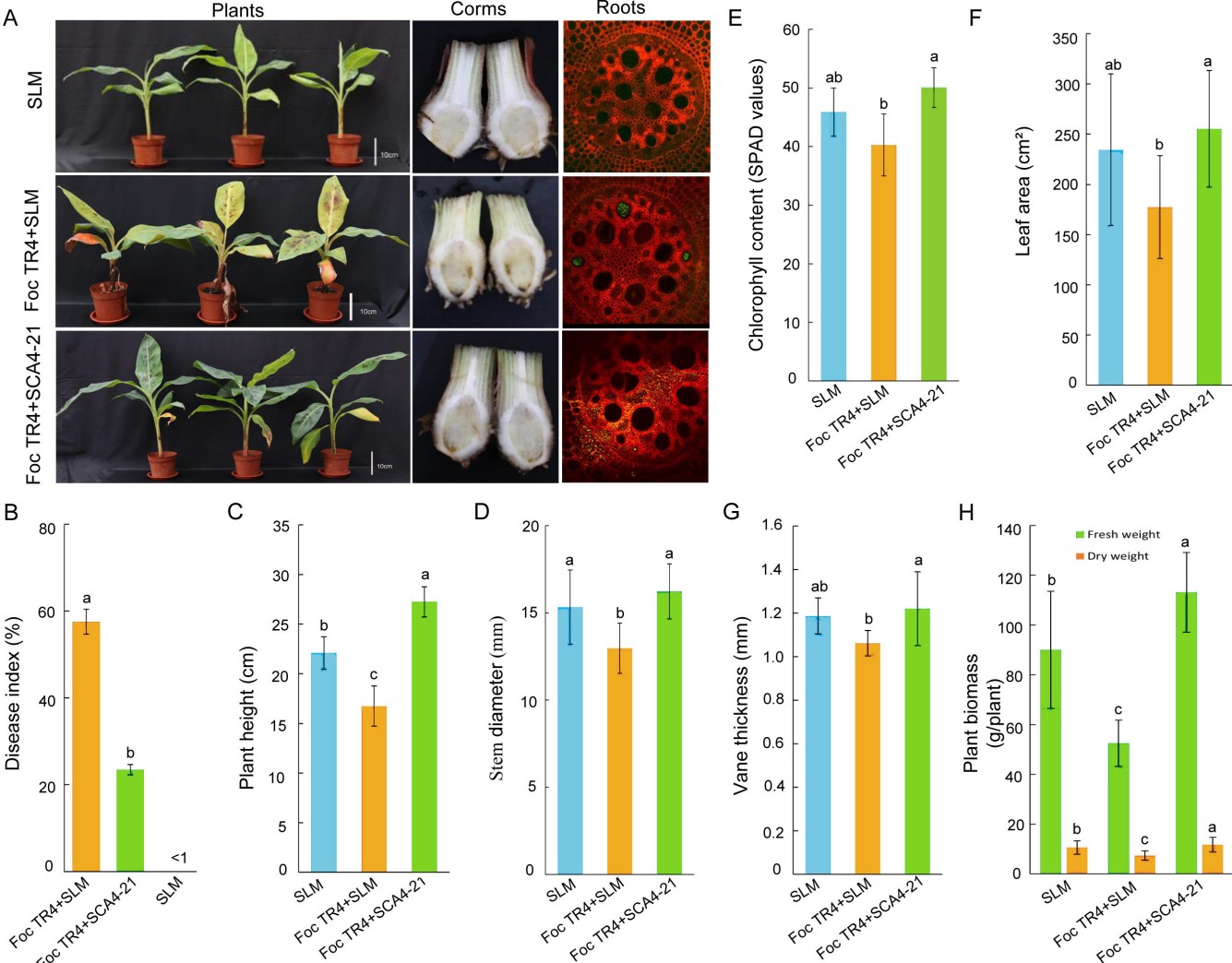

**FIG 4** *S. luomodiensis* SCA4-21 suppresses *Fusarium* wilt and promotes plant growth in banana seedlings. Plants were subjected to three treatments: SLM (control), Foc TR4 + SLM (inoculated with Foc TR4 and SLM), and Foc TR4 + SCA4-21 (co-inoculated with Foc TR4 and the biocontrol strain SCA4-21). (A) Biocontrol treatment alleviates disease symptoms and restricts fungal colonization. Representative images from 120 days post-inoculation include whole plants, corm cross-sections, and confocal laser scanning microscopy of vascular tissues (fungal hyphae in green). (B) Disease index. (C–H) Plant growth parameters: leaf chlorophyll content (E; SPAD value), leaf area (F), and leaf thickness (G) were all determined using the second fully expanded leaf. Additional parameters included (C) plant height, (D) stem diameter, and (H) plant biomass (fresh weight and dry weight). Data are presented as means ± SD ($n = 30$ plants). Different lowercase letters indicate significant differences ($P < 0.05$, Duncan's test).

some intra-group dispersion was observed, there was a clear separation along PC1: the bacterial community of the Foc TR4 + SCA4-21 group formed a distinct cluster, significantly separated from the SLM and Fcc TR4 + SLM groups, which showed closer proximity and partial overlap with each other. This indicates that the Foc TR4 + SCA4-21 treatment had a unique and pronounced effect on shaping the bacterial community. Nonmetric multidimensional scaling (NMDS) analysis result clearly showed significant variations in bacterial community composition across the different treatments (ANOSIM, $R = 0.909$, $P = 0.003$) and low stress value (stress = $0.0017 < 0.05$), indicating that the analysis results have excellent representativeness (Fig. 5H).

Taxonomic annotation of the bacterial communities across all samples yielded a total of 11,210 ASVs, spanning a wide range of phylogenetic ranks (Table 2). Analysis of the community composition at the class level revealed Gammaproteobacteria, Alphaproteobacteria, and Acidobacteria as the dominant taxa across all treatments (Fig. 6A). The Foc

**TABLE 1** Statistics of bacterial 16S rRNA gene sequencing data processing results for soil samples with different treatments

| Sample ID | Raw reads | Clean reads | Denoised reads | Merged reads | Non-chimeric reads |
|---|---|---|---|---|---|
| SLM | 79,966 | 79,771 | 74,069 | 59,601 | 50,140 |
| SLM | 80,121 | 79,953 | 74,192 | 58,275 | 47,570 |
| SLM | 79,926 | 79,767 | 74,365 | 60,998 | 51,339 |
| Foc TR4 + SLM | 80,039 | 79,878 | 74,308 | 66,628 | 59,763 |
| Foc TR4 + SLM | 80,123 | 79,937 | 73,295 | 60,729 | 51,707 |
| Foc TR4 + SLM | 79,909 | 79,735 | 72,242 | 59,815 | 51,019 |
| Foc TR4 + SCA4-21 | 80,362 | 80,187 | 74,852 | 68,004 | 58,432 |
| Foc TR4 + SCA4-21 | 79,863 | 79,684 | 74,327 | 70,695 | 62,599 |
| Foc TR4 + SCA4-21 | 79,878 | 79,682 | 73,340 | 62,188 | 49,566 |
| Total | 720,187 | 718,594 | 664,990 | 566,933 | 482,135 |

TR4 + SCA4-21 (T3) treatment was characterized by a notably higher relative abundance of Gammaproteobacteria and Bacilli compared to the other two treatments. The relative abundances of the top 20 bacterial genera are detailed in Table S3 and visualized in a heatmap overlaid with ANCOM-BC significance results (Fig. 6B). Compared to the T1 and T2 groups, the abundances of *Streptomyces, Bacillus, Sphingomonas, Massilia, Cupriavidus, Pseudomonas*, and *Burkholderia-Caballeronia-Paraburkholderia* were significantly increased in the T3 group. This specific enrichment highlights a key structural shift in the T3-associated microbial community. Conversely, the T3 group showed a significant decrease in the abundances of unclassified-*Acidobacteriales,* unclassified-SBR1031, unclassified-Xanthobacteraceae, *Bryobacter, Pantoea,* unclassified-*Elsterales,* unclassified-*Alphaproteobacteria*, and uncultured-*Acidobacteria-bacterium*.

Bacterial functional profiles based on the Kyoto Encyclopedia of Genes and Genomes (KEGG) were predicted via Tax4Fun (Fig. 7A). Compared to groups T1 and T2, the T3

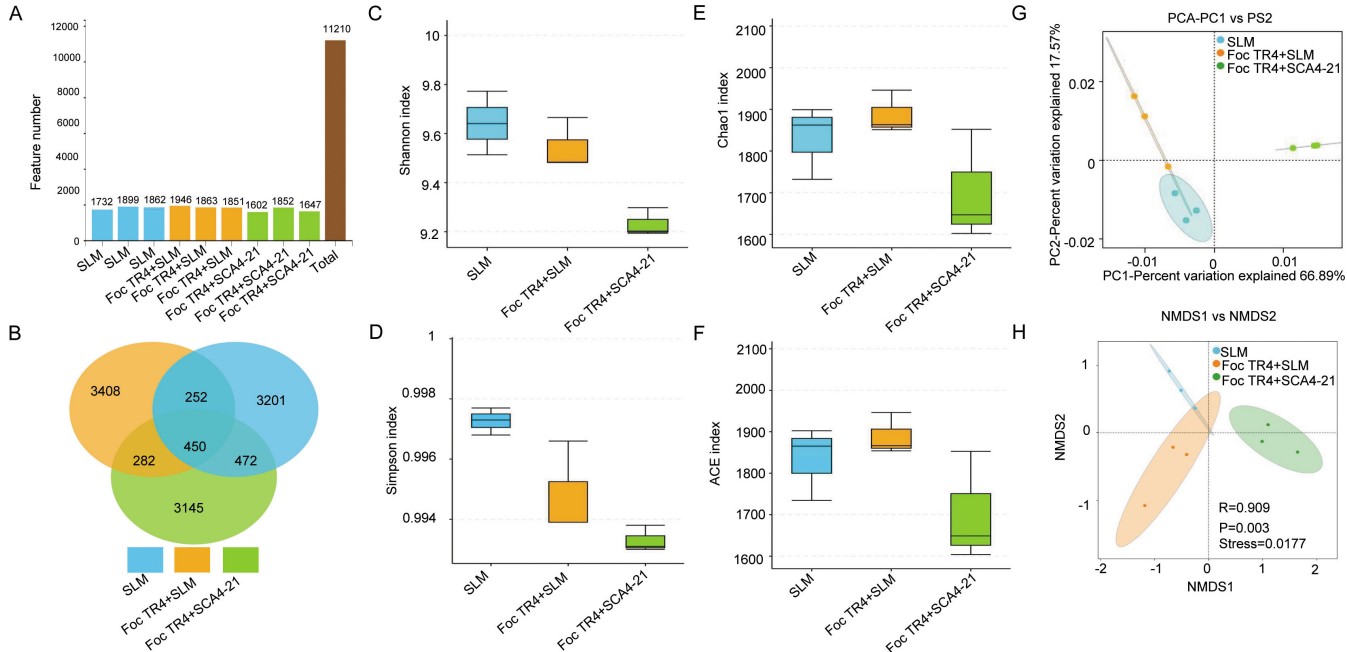

**FIG 5** *S. luomodiensis* SCA4-21 reshapes the structure but not the alpha diversity of the rhizosphere bacterial community. Rhizosphere soils were collected from three treatments: SLM (control), Foc TR4 + SLM (inoculated with Foc TR4 and SLM), and Foc TR4 + SCA4-21 (co-inoculated with Foc TR4 and the biocontrol strain SCA4-21). (A) Number of amplicon sequence variants. (B) Venn diagram of shared ASVs. (C and D) Boxplots of Shannon and Simpson diversity indices. (E and F) Boxplots of Chao1 and ACE richness indices. No significant differences were observed in these alpha-diversity indices among treatments ($P > 0.05$, Wilcoxon test). (G) Principal component analysis (PCA) and (H) non-metric multidimensional scaling (NMDS) ordination (Bray-Curtis distance; stress = 0.0177). ANOSIM confirmed significant differences in overall community structure among groups ($R = 0.909$, $P = 0.003$).

**TABLE 2** Bacterial species identified at different levels in soil samples under different treatments

| Sample | Kingdom | Phylum | Class | Order | Family | Genus | Species |
|---|---|---|---|---|---|---|---|
| SLM | 2 | 33 | 73 | 191 | 344 | 533 | 576 |
| SLM | 1 | 26 | 67 | 157 | 282 | 430 | 457 |
| SLM | 1 | 27 | 63 | 167 | 290 | 428 | 459 |
| Foc TR4 + SLM | 1 | 28 | 61 | 155 | 265 | 417 | 456 |
| Foc TR4 + SLM | 2 | 33 | 70 | 176 | 312 | 526 | 581 |
| Foc TR4 + SLM | 1 | 27 | 58 | 145 | 250 | 383 | 420 |
| Foc TR4 + SCA4-21 | 1 | 27 | 58 | 155 | 252 | 374 | 407 |
| Foc TR4 + SCA4-21 | 1 | 28 | 63 | 161 | 269 | 408 | 431 |
| Foc TR4 + SCA4-21 | 1 | 27 | 66 | 167 | 270 | 394 | 425 |
| Total | 2 | 40 | 99 | 265 | 515 | 971 | 1,142 |

group exhibited significant enrichment in pathways related to carbohydrate metabolism, amino acid metabolism, lipid metabolism, metabolism of terpenoids and polyketides, and biosynthesis of other secondary metabolites (Fig. 7A, I and II). Conversely, when comparing T1 and T3 groups, pathways such as nucleotide metabolism and translation were significantly increased in group T2 (Fig. 7A, I and III). Additionally, pathways such as folding, sorting and degradation, and replication and repair were significantly increased in both T2 and T3 groups compared to group T1 (Fig. 7A, II and III).

In order to explore the competitive or cooperative relationships between *Streptomyces* and other bacterial genera in differently treated soil samples, a co-occurrence network of the top 20 bacterial genera was constructed based on significant Spearman's rank correlations ($|\rho| > 0.1$, $P < 0.05$). The results revealed that the relative abundance of *Streptomyces* was significantly positively correlated with *Sphingomonas*, *Burkholderia-Caballeronia-Paraburkholderia*, *Massilia*, *Pseudomonas*, *Bacillus*, and *Cupriavidus*, while it

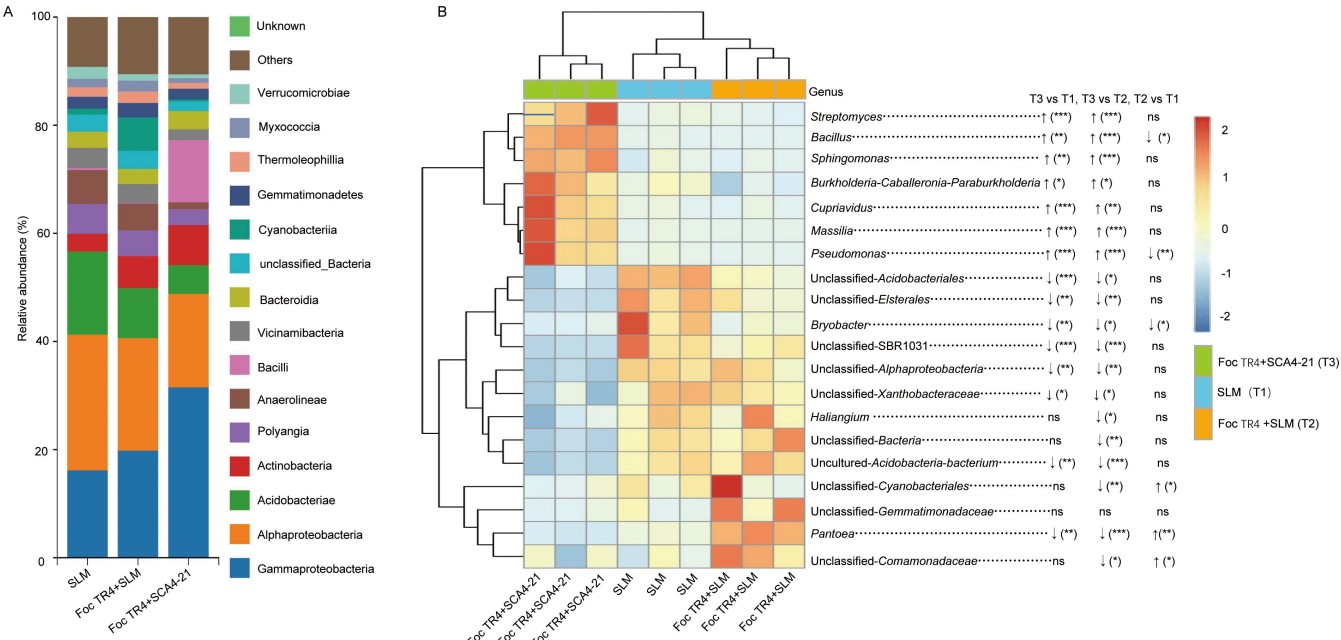

**FIG 6** *S. luomodiensis* SCA4-21 enriches the rhizosphere with beneficial bacteria and alters its taxonomic composition. (A) Relative abundance (based on ASV counts) of the top 15 bacterial classes, ranked by mean abundance across all samples. The biocontrol treatment (Foc TR4 + SCA4-21) was characterized by a notable increase in Gammaproteobacteria and Bacilli compared to the control (SLM) and pathogen-only (Foc TR4 + SLM) treatments. (B) Heatmap showing Z-score-scaled abundance of the top 20 most abundant genera. Differentially abundant genera were identified by ANCOM-BC analysis (corrected $P < 0.05$). Significance levels in figures are indicated as *$P < 0.05$, **$P < 0.01$, ***$P < 0.001$, and ns (not significant). The biocontrol treatment (T3) led to a significant enrichment of beneficial genera (e.g., *Streptomyces*, *Bacillus*, *Sphingomonas*, *Pseudomonas*, and *Massilia*) and a reduction in unclassified taxa (e.g., *Pantoea*).

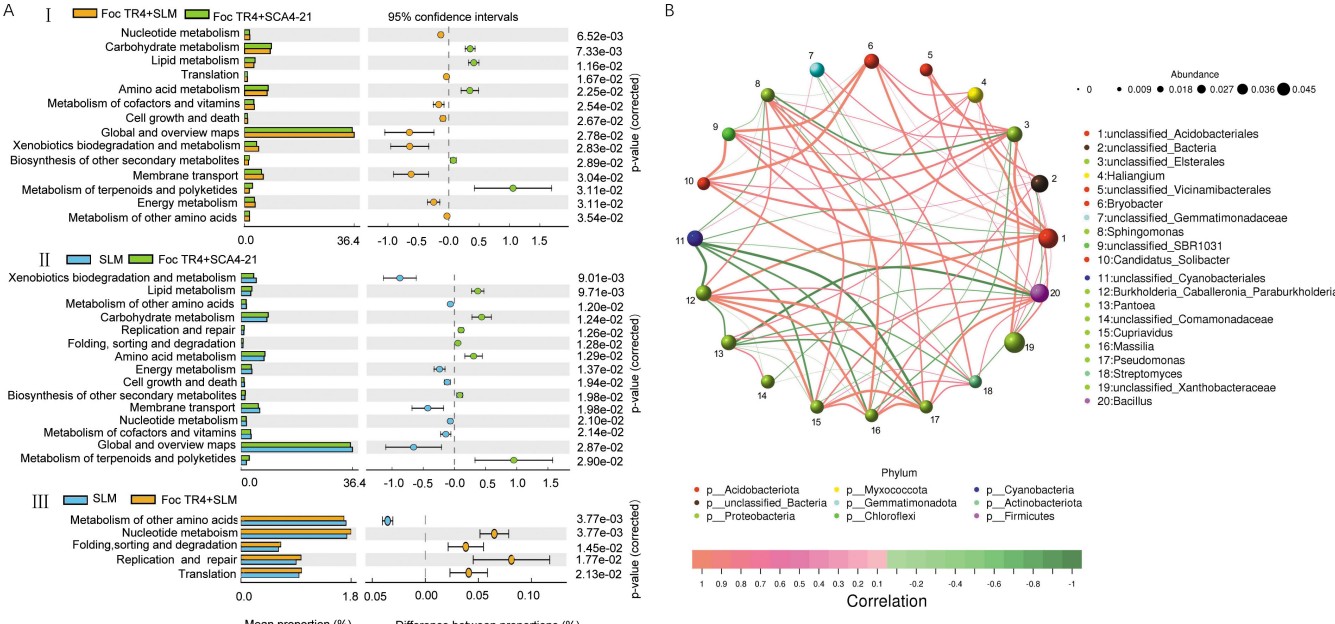

**FIG 7** *S. luomodiensis* SCA4-21 enriches metabolic pathways and builds a synergistic network in the rhizosphere. (A) Pairwise comparisons of KEGG pathways (level 2) predicted by Tax4Fun2. The panels represent the following comparisons: I, Foc TR4 + SLM (T2) vs Foc TR4 + SCA4-21 (T3); II, SLM (T1) vs Foc TR4 + SCA4-21 (T3); III, SLM (T1) vs Foc TR4 + SLM (T2). The biocontrol group (T3) showed significant enrichment in key metabolic pathways (e.g., carbohydrate and secondary metabolite biosynthesis) compared to the pathogen (T2) and control (T1) groups, as analyzed by Welch's *t*-test ($P < 0.05$). (B) Co-occurrence network of the top 20 bacterial genera, constructed based on significant Spearman correlations ($|\rho| > 0.1$, $P < 0.05$). Analysis revealed that *Streptomyces* (highlighted) formed significant positive correlations with multiple beneficial genera (e.g., *Sphingomonas*, *Pseudomonas*, and *Bacillus*), indicating a synergistic consortium.

was significantly negatively correlated with *Pantoea* and unclassified *Cyanobacteriales* (Fig. 7B).

## Strain SCA4-21 monitors the structure of the banana rhizosphere fungal community

Fungal ITS analysis of nine soil samples yielded 720,798 pairs of raw reads. Following splicing, chimera removal, and denoising procedures, we obtained a set of high-quality clean reads comprising 690,176 pairs (Table 3). These authentic biological sequences were subsequently clustered into 2,454 fungal ASVs. Specifically, the three soil samples from the T1 group contributed 309, 411, and 590 ASVs, respectively. The three soil samples from the T2 group comprised 316, 446, and 429 ASVs, respectively. The three soil samples from the T3 group generated 413, 341, and 480 ASVs, respectively (Fig. 8A). Venn

**TABLE 3** Statistics of fungal internal transcribed spacer sequencing data processing results for soil samples with different treatments

| Sample ID | Raw reads | Clean reads | Denoised reads | Merged reads | Non-chimeric reads |
|---|---|---|---|---|---|
| SLM | 79,906 | 79,566 | 78,867 | 77,548 | 75,255 |
| SLM | 79,890 | 79,505 | 78,644 | 77,148 | 77,023 |
| SLM | 80,044 | 79,637 | 78,167 | 76,089 | 74,962 |
| Foc TR4 + SLM | 80,085 | 79,814 | 79,456 | 78,027 | 77,692 |
| Foc TR4 + SLM | 80,424 | 80,067 | 79,277 | 77,274 | 77,216 |
| Foc TR4 + SLM | 80,110 | 79,831 | 79,108 | 76,998 | 76,793 |
| Foc TR4 + SCA4-21 | 80,324 | 79,973 | 79,605 | 77,926 | 77,814 |
| Foc TR4 + SCA4-21 | 79,984 | 79,581 | 79,301 | 78,148 | 77,432 |
| Foc TR4 + SCA4-21 | 80,031 | 79,698 | 79,056 | 76,523 | 75,989 |
| Total | 720,798 | 717,672 | 711,481 | 695,681 | 690,176 |

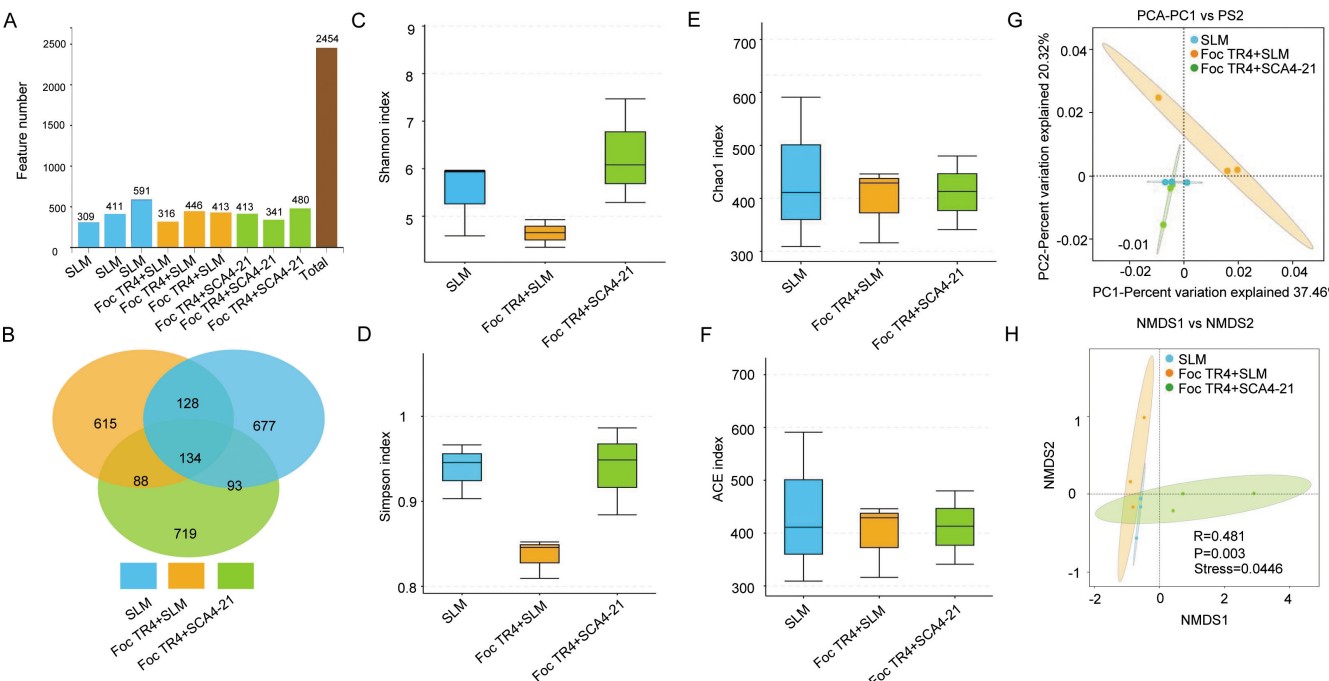

**FIG 8** *S. luomodiensis* SCA4-21 reshapes the structure of the rhizosphere fungal community without altering its alpha diversity. Rhizosphere soils were collected from three treatments: SLM (control), Foc TR4 + SLM (pathogen), and Foc TR4 + SCA4-21 (biocontrol). (A) Number of fungal ASVs. (B) Venn diagram of shared ASVs. (C and D) Boxplots of Shannon and Simpson diversity indices. (E and F) Boxplots of Chao1 and ACE richness indices. No significant differences were observed in these alpha-diversity indices among treatments ($P > 0.05$, Wilcoxon test). (G) Principal component analysis and (H) non-metric multidimensional scaling ordination (Bray-Curtis distance; stress = 0.0446) show a clear separation of fungal community structures across treatments. ANOSIM confirmed significant differences in overall community structure among groups ($R = 0.481$, $P = 0.003$).

analysis demonstrated that T1, T2, and T3 groups had 677, 615, and 719 unique fungal ASVs, respectively (Fig. 8B).

Analysis of fungal alpha diversity indicated that the diversity indices Shannon and Simpson in T3 group soils were higher than those in T1 and T2 group soils (Fig. 8C and D), while the richness indices Chao1 and ACE were slightly higher in T2 group soils than in T1 and T3 group soils (Fig. 8E and F). However, no statistically significant differences were observed in the fungal diversity indices Shannon and Simpson, as well as the richness indices ACE and Chao1, among the three treatments (Fig. 8C through F; Table S4).

The fungal community structure, visualized by PCA, demonstrated that the Foc TR4 + SLM treatment led to a uniquely high level of community heterogeneity, with its replicates widely scattered across the ordination space. Microbial communities under the SLM and Foc TR4 + SCA4-21 treatments clustered closely together in the lower-left quadrant, indicating they were highly stable and similar. Only slight sample dispersion within this shared cluster suggested minor intra-group variability. This clear contrast underscores that the Foc TR4 + SLM condition was the primary driver of unstable and divergent fungal assemblies (Fig. 8G). The NMDS analysis revealed significant differences in fungal community composition among the different treatments (ANOSIM, $R = 0.481$, $P = 0.003$) with a low stress value (stress = 0.0446 < 0.05), suggesting the highly representative nature of the analysis results (Fig. 8H).

Taxonomic annotation of the fungal communities across all samples identified a total of 2,454 ASVs, distributed across various phylogenetic ranks as detailed in Table 4. The overall composition at the class level is shown for the top 15 taxa in Fig. 9A. The communities across all groups were predominantly composed of Sordariomycetes, Agaricomycetes, and unclassified Basidiomycota. A notable observation was the substantially higher relative abundance of Sordariomycetes in the T2 Group compared to the other groups. Conversely, T3 Group exhibited a marked increase in the proportion of

**TABLE 4** Fungal species identified at different levels in soil samples under different treatments

| Sample | Kingdom | Phylum | Class | Order | Family | Genus | Species |
|---|---|---|---|---|---|---|---|
| SLM | 1 | 9 | 24 | 53 | 85 | 106 | 111 |
| SLM | 1 | 8 | 26 | 59 | 93 | 133 | 144 |
| SLM | 1 | 10 | 32 | 72 | 132 | 200 | 227 |
| Foc TR4 + SLM | 1 | 8 | 24 | 59 | 92 | 131 | 142 |
| Foc TR4 + SLM | 1 | 10 | 29 | 69 | 111 | 139 | 142 |
| Foc TR4 + SLM | 1 | 10 | 27 | 61 | 97 | 129 | 136 |
| Foc TR4 + SCA4-21 | 1 | 11 | 30 | 70 | 110 | 166 | 196 |
| Foc TR4 + SCA4-21 | 1 | 9 | 29 | 54 | 98 | 140 | 160 |
| Foc TR4 + SCA4-21 | 1 | 10 | 26 | 65 | 120 | 173 | 189 |
| Total | 1 | 11 | 42 | 105 | 217 | 400 | 537 |

Mortierellomycetes and Tremellomycetes. The relative abundances of the top 20 fungal genera are presented in Table S5 and a corresponding heatmap incorporating ANCOM-BC significance results (Fig. 9B). The *Fusarium* abundance in T2 was significantly higher than in T1. Consequently, the T3 group exhibited an intermediate level of *Fusarium*, which was not significantly different from either the elevated T2 group or the basal T1 group. The T3 treatment also led to significant changes in other fungal genera. The abundances of *Gibellulopsis*, *Xenomyrothecium*, and *Mortierella* were significantly elevated in T3 compared to both the T1 and T2 groups. Furthermore, genera, including *Purpureocillium*, *Conocybe*, *Lycoperdon*, *Cladosporium,* and *Arthrobotrys,* also displayed clear increasing trends in the T3 soils. Conversely, several taxa that were prominent in the T1 or T2 soils were markedly reduced in the T3 group. These included *Rhizophagus*, unclassified *Agaricomycetes*, unclassified *Sordariomycetes*, unclassified *Dothideomycetes*, unclassified G*lomeraceae*, and *Gymnopilus*.

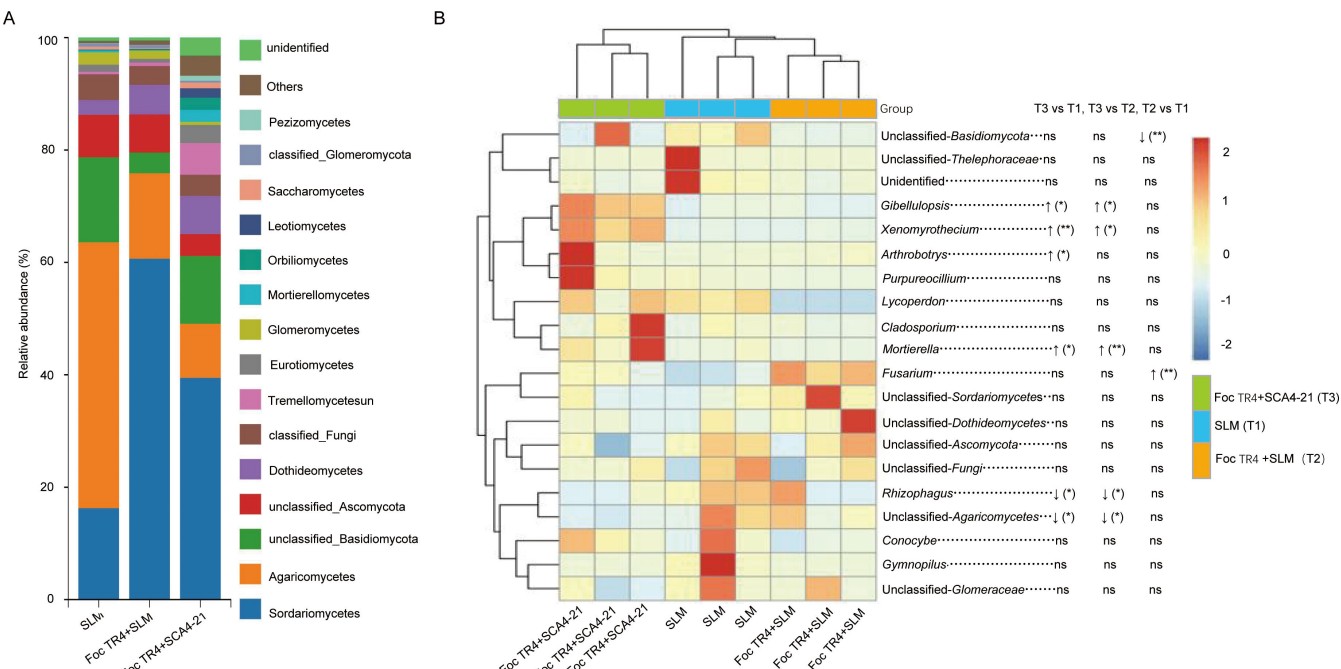

**FIG 9** *S. luomodiensis* SCA4-21 restructures the rhizosphere fungal composition by suppressing *Fusarium* and enriching beneficial taxa. (A) Relative abundance (based on ASV counts) of the top 15 fungal classes. The biocontrol group (T3, Foc TR4 + SCA4-21) was characterized by a marked increase in Mortierellomycetes and Tremellomycetes, while the pathogen group (T2, Foc TR4 + SLM) was dominated by Sordariomycetes. (B) Heatmap showing *Z*-score-scaled abundance of the top 20 most abundant genera. ANCOM-BC analysis revealed that the T3 treatment reduced the abundance of *Fusarium* to an intermediate level and promoted the enrichment of beneficial genera such as *Gibellulopsis*, *Mortierella*, and *Purpureocillium*. Significance levels in figures are indicated as *$P < 0.05$, **$P < 0.01$, ***$P < 0.001$, and ns (not significant).

## DISCUSSION

### Antifungal activity and volatile metabolites of strain SCA4-21

*Streptomyces* spp. are a significant source of microbial biocontrol agents, primarily through the production of antimicrobial bioactive substances. Behera et al. (20) demonstrated that *S. chilikensis* RC1830 exhibited *in vitro* antifungal activity against *F. oxysporum* through metabolite production, reducing rice wilt severity by 80.51% in pot experiments. Chen et al. (21) also reported that *S. plicatus* B4-7 synthesized the antifungal compound borrelidin, reducing crown rot incidence by 75%. Our previous studies confirmed that *Streptomyces* sp. CB-75 and YYS-7 significantly decreased banana *Fusarium* wilt incidence by inhibiting mycelial growth, spore germination, and disrupting the cell ultrastructure of Foc TR4 (16, 22).

In this study, the extract of *S. luomodiensis* SCA4-21 significantly inhibited Foc TR4 hyphal growth (Fig. 1A and B) and spore germination (Fig. 2A and B), severely disrupting hyphal and spore ultrastructure (Fig. 2C through E). The extract also exhibited broad-spectrum antifungal activity against eight other phytopathogenic fungi (Fig. 1C). GC-MS analysis identified 32 volatile organic compounds produced by strain SCA4-21 (Fig. 3; Table S1), including methyl butanoate, methyl 3-methylbutanoate, 5-methyl-2-heptanone, 6-methyl-2-heptanone, 3-octanone, methyl 2-ethylhexanoate, and methyl benzoate. Crucially, several of these VOCs have been previously reported as potent antifungal agents. For instance, 5-methyl-2-heptanone and 6-methyl-2-heptanone show potent activity against *Penicillium italicum* and *Alternaria solani*, respectively (23, 24). Similarly, methyl 2-ethylhexanoate and methyl benzoate exhibit significant antifungal activity against *Colletotrichum gloeosporioides* and *Candida albicans* (25, 26), suggesting they are primary contributors to the antifungal activity of strain SCA4-21. The VOC profile also included compounds with other biological activities: 3-octanone promotes plant growth in *Arabidopsis thaliana* (27), methyl butanoate has anticancer activity (28), and methyl 3-methylbutanoate has nematicidal activity (29). This diverse blend of bioactive volatiles highlights the potential of strain SCA4-21 for direct pathogen inhibition and enhanced plant health.

### Biocontrol efficacy and plant growth promotion in pot experiments

The diverse bioactive volatiles produced by strain SCA4-21 contributed to its high biocontrol efficiency against banana *Fusarium* wilt in pot experiments. It suppressed Foc TR4 mycelial infection in roots, achieving a remarkable 59.3% reduction in disease incidence and significantly promoted banana seedling growth (Fig. 4A through H). Biocontrol efficacy often involves both direct antagonism and modulation of the resident rhizosphere microbiome, which serves as the first defense against soil-borne pathogens (30). Therefore, we investigated whether strain SCA4-21 inoculation reshaped the microbial community structure.

### Reshaping of the rhizosphere bacterial community

Our high-throughput sequencing results confirmed this restructuring. Comparative analysis revealed that the T3 group (inoculated with Foc TR4 + SCA4-21) exhibited a significant increase in the abundances of several bacterial genera, including *Streptomyces*, *Bacillus*, *Sphingomonas*, *Massilia*, *Pseudomonas*, *Burkholderia-Caballeronia-Paraburkholderia*, and *Cupriavidus*, compared to T1 (SLM control) and T2 (Foc TR4 + SLM) groups (Fig. 6B). Conversely, a significant decrease in the abundance of the genus *Pantoea* was observed in the T3 group compared to the T1 and T2 groups (Fig. 6B). Co-occurrence network analysis further revealed significant positive correlations between *Streptomyces* and the enriched genera (e.g., *Bacillus*, *Sphingomonas*, *Pseudomonas*, *Massilia*, *Burkholderia-Caballeronia-Paraburkholderia*, and *Cupriavidus*) and a significant negative correlation with the suppressed genus *Pantoea* (Fig. 7B). While correlation does not equate to causation, this network topology allows us to hypothesize about potential microbial

interactions: the positive correlations may indicate cooperative or commensal relationships, while the negative correlation with *Pantoea* is highly suggestive of a competitive or antagonistic relationship, which is consistent with the well-documented role of *Streptomyces* as prolific producers of antimicrobial compounds. These inferred relationships require further validation through future studies, such as co-culture assays or meta-transcriptomic analysis.

The accumulated data strongly support the beneficial roles of several key enriched genera. Specifically, *Bacillus*, *Sphingomonas*, and *Pseudomonas* are well documented as cornerstone biological control agents and are frequently recruited by microbial inoculants to defend against soil-borne diseases (31–35). For instance, Zhang et al. (36) and Yang et al. (37) reported that biocontrol agents triggered plant resistance or suppressed *Fusarium* wilt by enriching *Pseudomonas* and *Sphingomonas* or attracting *Bacillus*. Our previous studies also confirmed the significant enrichment of these genera in the healthy banana rhizosphere (38). Beyond these well-established beneficial taxa, the consortium was complemented by other enriched genera. For example, *Massilia* is a common rhizosphere colonizer, with some isolates known to control pathogens via siderophores or extracellular enzymes (39–42). Similarly, while the direct biocontrol applications of *Burkholderia-Caballeronia-Paraburkholderia* and *Cupriavidus* are less documented, they are frequently identified as beneficial dominants in various niches (37). Conversely, the significant suppression of the genus *Pantoea*—which contains species frequently recognized as plant pathogens (43, 44)—aligns with findings that biocontrol agents can reduce its abundance in diseased soils (45).

## Functional reprogramming of the bacterial microbiota

The successful assembly of this beneficial bacterial consortium, complemented by the suppression of a pathogenic genus, prompted us to investigate whether this structural reshaping was reflected in the community's metabolic potential. The Tax4Fun prediction, which is based on 16S rRNA gene data from the bacterial microbiota, revealed that the T3 group underwent a profound functional reprogramming characterized by the significant upregulation of key metabolic pathways compared to the T1 and T2 groups (Fig. 7A). These included pathways for the biosynthesis of other secondary metabolites, carbohydrate metabolism, lipid metabolism, and amino acid metabolism (Fig. 7A, I and II). This functional signature indicates that strain SCA4-21 reshaped a microbiome with a heightened capacity for producing antimicrobial compounds while simultaneously enhancing its fundamental metabolic activities linked to nutrient turnover and plant growth promotion (46).

## Modulation of the fungal community

In parallel to the bacterial community shifts, our findings also revealed the impact of strain SCA4-21 on the fungal community. Crucially, the introduction of the pathogen Foc TR4 (T2 group) significantly enhanced the abundance of *Fusarium*. The subsequent inoculation of strain SCA4-21 (T3 group) counteracted this increase, restricting the *Fusarium* population to a level that was not significantly different from the healthy control (T1) (Fig. 9B). This suggests that strain SCA4-21 limited the pathogen's establishment, an effect that is likely supported by the concurrently observed and significant restructuring of the protective bacterial community. This aligns with the established understanding that the severity of banana *Fusarium* wilt is directly linked to increased *Fusarium* colonization in the field (38) and that successful biocontrol agents often act by suppressing *Fusarium* growth in the soil, as demonstrated by *Bacillus amyloliquefaciens* NJN-6 (47). In addition to the pathogen suppression, strain SCA4-21 also promoted the enrichment of several beneficial fungal genera (Fig. 9B). We observed that the relative abundances of *Mortierella*, *Gibellulopsis*, and *Xenomyrothecium* were significantly higher in the T3 group soils, with the abundance of *Mortierella* being particularly striking (approximately 9 and 33 times higher than in T1 and T2 groups, respectively; Table S5). Additionally, the genus *Purpureocillium* also showed an increasing trend in the T3

group. The enrichment of these taxa is likely indicative of a disease-suppressive soil state. The genus *Mortierella* is frequently recognized as a beneficial fungus with antifungal activity (48, 49). Its decline is associated with soil degradation and pathogen increase (50), while its predominance, as we previously documented in disease-free soils (38), is a hallmark of soil health. Similarly, *Purpureocillium* is a well-documented biocontrol agent against nematodes (51) and can promote plant growth (52), and its increasing trend here aligns with this beneficial role. The role of *Gibellulopsis* is complex; while it contains pathogenic species, numerous hypovirulent strains are employed as biocontrol agents against *Verticillium* wilt (53–55), and its increase has been linked to beneficial crop rotation practices (56). Finally, *Xenomyrothecium*, primarily a saprophytic genus crucial for decomposition, also includes endophytic species with significant antagonistic activity against pathogens (57, 58).

In conclusion, we propose a dual mechanism for the biocontrol efficacy of strain SCA4-21 against banana *Fusarium* wilt. First, it acts directly against Foc TR4, likely through the production of antifungal compounds. Second, it indirectly suppresses the pathogen by reshaping the soil microbiome into a disease-suppressive state. This is achieved by selectively enriching beneficial bacteria (e.g., *Bacillus* and *Pseudomonas*) and fungi (e.g., *Mortierella*), while simultaneously suppressing pathogenic competitors like *Pantoea* and the *Fusarium* pathogen itself. This synergistic interplay between direct antagonism and microbiome-mediated resistance represents a robust strategy for sustainable disease management. It should be noted that these findings were demonstrated under controlled pot conditions. Future work is essential to validate this efficacy in field trials and develop stable formulations. Given this promising foundation and its elucidated mechanisms, strain SCA4-21 presents a strong candidate for development into commercial biocontrol products, such as soil amendments or seedling treatments, for sustainable banana production.

## Conclusion

*S. luomodiensis* SCA4-21 demonstrates great potential as a biocontrol agent against banana *Fusarium* wilt. Its efficacy is achieved through a dual mechanism: (i) direct antagonism via the production of diverse antifungal volatile compounds that disrupt the pathogen Foc TR4; and (ii) indirect mediation by profoundly reshaping the rhizosphere microbiome. This reshaping enriches a consortium of beneficial bacteria (e.g., *Bacillus* and *Sphingomonas*) and fungi (e.g., *Mortierella*), while suppressing pathogenic populations (*Fusarium* and *Pantoea*), ultimately fostering a disease-suppressive soil environment. Our findings highlight strain SCA4-21 as a multifaceted and sustainable solution for managing this devastating disease.

## MATERIALS AND METHODS

### Extraction and purification of strain SCA4-21 extract

The extract of strain SCA4-21 was obtained and purified following the method of Li et al. (59). Namely, strain SCA4-21 was inoculated into 1,000 mL of SLM (20 g of soluble starch, 15 g of soybean powder, 5 g of yeast powder, 2 g of peptone, 4 g of CaCO$_3$, 4 g of NaCl, and 1 L of ddH$_2$O, pH 7.2–7.4) and cultured at 28°C for 7 days at 180 r/min. The fermentation broth was mixed with an equal volume of 95% ethanol and then shaken at 28°C and 180 rpm for 48 h. The mixtures were filtered by filter paper, and the filtrate was concentrated by rotary vacuum evaporator (N-1300, BoLang Co., Ltd., Shanghai, China). To separate and purify the extract, the concentrate was fully adsorbed onto macroporous resin (D101) and eluted sequentially with 50%, 60%, 70%, 80%, and 100% methanol gradient. These eluents were respectively evaporated under reduced pressure to yield syrupy residues. Each residue was dissolved in 10% dimethyl sulfoxide at a concentration of 20 mg/mL, filtered through a 0.22 µm sterile filter (Millipore, Bedford, MA, USA), and stored at −4 °C.

The antifungal activity of the extract against Foc TR4 was determined using agar dilution method (60). Briefly, 100 µL of extract solutions from different eluents was added to 60 mL of potato dextrose agar (PDA) medium that had been sterilized but not yet solidified, respectively. Every 60 mL of potato dextrose agar medium is evenly poured into three petri dishes to make plates. Negative controls consisted of PDA amended with 10% (vol/vol) DMSO. The positive control was PDA amended with tebuconazole (33.3 µg/mL) dissolved in 10% (vol/vol) DMSO. All treatments were performed in biological triplicate. Each plate in the center was inoculated with a Foc TR4 mycelial plug with a 0.5 cm diameter and incubated at 28°C for 7 days. The colonial diameter of Foc TR4 was measured, and fungal growth inhibition (FGI) was calculated using the following formula: FGI $= [(C - T)/C] \times 100\%$, where $C$ and $T$ represent the growth diameters of the control and the treatment group, respectively. The most effective extract was selected for further investigations.

## Measuring the $EC_{50}$ of strain SCA4-21 extract

PDA plates with final concentrations of extract at 0.78, 1.56, 3.13, 6.25, 12.50, 25.00, 50.00, 100.00, and 200.00 µg/mL, respectively, were prepared. The fungal disk of Foc TR4 (0.5 cm diameter) was inoculated on PDA plates and cultivated in an incubator at 28°C for 7 days. The plates with DMSO (10%, vol/vol) were used as a control. Three biological replicates for each concentration were performed. The growth diameter of Foc TR4 was measured, and the percentage of mycelial inhibition was calculated. A linear regression equation was established using the least squares method to determine the half-maximal effective concentration ($EC_{50}$) of extract on the mycelial growth of Foc TR4 (16).

## Determining the broad-spectrum antifungal activity of strain SCA4-21 extract

To assess the potential application of strain SCA4-21 extract against a wider range of economically significant crop diseases, the broad-spectrum antifungal activity of the extract was determined against a panel of eight fungal isolates using the agar well diffusion method (61). Specifically, a mycelial plug (5 mm in diameter) taken from the actively growing margin of a 7-day-old fungal colony was inoculated at the center of a PDA plate. Using a sterile cork borer, wells (5 mm in diameter) were aseptically punched into the agar at a distance of 2.5 cm from the central inoculum. A volume of 100 µL of filter-sterilized extract (100 µg/mL) was added into each well. An equal volume of 10% DMSO served as the control. All experiments included three independent biological replicates. The plates were incubated at 28°C for 5–7 days. The antifungal activity was quantified by measuring the colonial diameter of the tested pathogen. The FGI rate was calculated according to the formula mentioned above.

The panel of fungal isolates comprised six species from three genera (*Curvularia, Colletotrichum,* and *Fusarium*) that cause devastating diseases in a wide range of economically important crops (e.g., cereals, fruits, and vegetables). The specific isolates tested were as follows: *Curvularia fallax* (ATCC 34598), *Colletotrichum gloeosporioides* (Penzig) Penzig et Saccardo (ATCC MYA-456), *C. gloeosporioides* (ACCC 36351), *C. acutatum* (ATCC 56815), *C. fragariae* (ATCC 58718), *C. gloeosporioides* (Penz) Saec (ACCC 36351), *F. oxysporum* f. sp. *cucumerinum* Owen. anamorph (ATCC 204378), and *F. graminearum* Sehw (DSM 21803). The pathogens were kindly provided by the Institute of Environment and Plant Protection, China Academy of Tropical Agricultural Sciences, Haikou, China.

## Effect of strain SCA4-21 extract on spore germination of Foc TR4

The effect of strain SCA4-21 extract on spore germination of Foc TR4 was observed using our previous method (62) with minor modifications. A volume of 5 mL of sterile water was added to the well-growing Foc TR4 plate, and the mycelia and spores were scraped using the sterilized coating rod. The spore suspension obtained by filtering off mycelia was diluted to $1 \times 10^6$ CFU/mL and mixed with equal volume of different concentrations

of the extract ($1\times$ EC$_{50}$, $2\times$ EC$_{50}$, $4\times$ EC$_{50}$, and $8\times$ EC$_{50}$). DMSO (10%) was used as a control instead of the extract. All experiments were performed in three independent biological replicates. After incubating at 28°C for 24 h, Foc TR4 spore germination was assessed using an inverted microscope (Cellcutplus, MMI, Germany). A total of 100 spores were counted per biological replicate (100 spores $\times$ 3 microscopic fields; $n = 3$ replicates). A spore was considered germinated when the length of its germ tube exceeded half of the spore's diameter. The spore germination inhibition (SGI) was calculated following the formula: SGI = $(A - B)/A \times 100\%$, where $A$ and $B$ signify the spore germination rate of the control group and the treatment group, respectively.

## Effect of strain SCA4-21 extract on mycelial and spore morphology as well as the ultrastructure of Foc TR4

The effect of strain SCA4-21 extract on mycelial morphology of Foc TR4 was observed following the method of Cao et al. (63), with some modifications. Three PDA plates containing $4\times$ EC$_{50}$ extract of strain SCA4-21 were utilized as treatments, while three PDA plates containing 10% DMSO served as controls. Disks (0.5 cm in diameter) of Foc TR4 were inoculated at the centers of these plates. After 3 days of culture, fungal mycelia blocks (0.5 cm diameter) were cut from the edge of the pathogen clone, fixed with 2.5% (vol/vol) glutaraldehyde (C$_3$H$_8$O$_2$) at −4°C overnight, and rinsed twice using phosphate-buffered saline (0.1 mol/L, pH 7.4). Then, those fungal mycelia blocks underwent gradient dehydration in 30%, 50%, 70%, 80%, 90%, 95%, and 100% ethanol for 2 min each, followed by a rinse with tert-butanol for 20 min. These fungal mycelium blocks were subsequently soaked in fresh tert-butanol and frozen at −80°C. They were then subjected to freeze-drying using a freeze dryer (FDU-2110, EYELA, Tokyo, Japan). The sample sheet was fixed on a small steel column and coated with a film of gold-palladium alloy under vacuum. Then, the morphology of the hypha was observed by scanning electron microscopy (Zeiss SIGMA, Germany). Fungal hyphae were also collected from the above 3-day-old treatment and control plates, respectively, and treated according to our previous method (62). The ultrastructure of fungal hyphae was detected by transmission electron microscopy (JEM-1400Flash, Japan).

In addition, Foc TR4 spore suspension with $1 \times 10^6$ CFU concentration was prepared according to the above method and mixed with $4\times$ EC$_{50}$ extract solution of strain SCA4-21 (vol:vol = 1:1). A volume of 20 µL of mixture was dropped onto a cover plate, which was placed on a glass slide and cultivated at 28°C for 24 h. Control was set up by replacing the extract solution with 10% DMSO. Spores were treated as described above and observed by SEM (Zeiss SIGMA, Germany).

## Identification of the volatile organic compounds of strain SCA4-21

The volatile organic compounds of strain SCA4-21 were analyzed using gas chromatography coupled with mass spectrometry (Clarus690 + SQ8T, PerkinElmer, USA) following the method described in our previous study (63). Specifically, a volume of 100 µL of strain SCA4-21 suspension ($10^6$ CFU mL$^{-1}$) was inoculated into 15 mL of sterile SLM medium in a 50 mL flask, with an equal volume of ddH$_2$O used as the control. The treatment and control groups were set up with three biological replicates each. All flasks were sealed with sterile foil and film and incubated at 28°C with shaking at 180 rpm for 7 days.

Volatiles were collected using an SPME fiber assembly (Supelco Co., PA, USA; 50/30 µm DVB/CAR/PDMS). The fiber was inserted through the septum into the headspace of the flask, which was maintained in a glycerol bath at 50 °C, to adsorb volatile compounds. The fiber was then thermally desorbed in the GC injector at 250°C. The GC was equipped with an ELITE-5MS capillary column (30 m $\times$ 0.25 mm $\times$ 0.25 µm; Agilent, USA). The oven temperature program was set as follows: initial temperature of 40°C (held for 1 min), increased to 160°C at a rate of 3°C min$^{-1}$, increased to 230°C at a rate of 15°C min$^{-1}$ and held for 2 min, and finally increased to 250°C at a rate of 10°C min$^{-1}$ and held for 5 min. Helium was used as the carrier gas at a constant flow rate of 1.0 mL min$^{-1}$. The mass spectrometer was operated in the electron impact mode at

70 eV, with the ion source temperature set to 250°C. The mass scan range was from $m/z$ 35 to 300. Compounds were initially identified by comparing their mass spectra with the NIST11 mass spectral library. The VOC profiles of strain SCA4-21 and the control were compared. This allowed us to pinpoint compounds that were unique to or had significantly higher peak areas in the treatment; these compounds were then identified by matching their retention indices to reference standards.

## Evaluation of the biocontrol efficiency of strain SCA4-21 and its promotion effect on banana seedlings

To evaluate the biocontrol efficacy of the live strain SCA4-21 under conditions that simulate a natural soil-plant environment, a pot experiment was conducted to evaluate the biocontrol potential of strain SCA4-21 against banana *Fusarium* wilt. The GFP-tagged Foc TR4 strain, provided kindly by the Institute of Tropical Bioscience and Biotechnology, China Academy of Tropical Agricultural Sciences (Haikou, China), was used. A spore suspension was prepared by inoculating the pathogen in potato dextrose broth and incubating at 28°C with shaking (180 rpm) for 7 days and adjusted to a final concentration of $1 \times 10^6$ CFU/mL.

Prior to planting, the roots of healthy banana seedlings (three to four leaf stage) in the positive control (T2) and treatment (T3) groups were wounded with scissors and immersed in the spore suspension of GFP-Foc TR4 for 30 min to facilitate infection. To serve as a proper negative control, seedlings in the T1 group underwent the same physical wounding and immersion procedure but in sterile water instead of the spore suspension. All seedlings were then planted in plastic pots containing 900 g of soil.

Three experimental groups were established: T1 (negative control): seedlings were wounded and immersed in sterile water, then watered weekly with 100 mL of a 10-fold diluted sterilized soybean liquid medium. T2 (positive control): seedlings were wounded and immersed in the GFP-Foc TR4 spore suspension, then irrigated weekly with 100 mL of the spore suspension ($1 \times 10^6$ CFU/mL) plus 100 mL of 10-fold diluted SLM. T3 (treatment): seedlings were wounded and immersed in the GFP-Foc TR4 spore suspension, then irrigated weekly with 100 mL of the spore suspension ($1 \times 10^6$ CFU/mL) plus 100 mL of a 10-fold diluted fermentation broth of strain SCA4-21.

The fermentation broth of strain SCA4-21 was prepared by inoculating the isolate into SLM and incubating at 28°C with shaking (180 rpm) for 7 days. Each treatment consisted of 30 pots (replicates).

After banana seedlings were planted in a greenhouse at 28°C, with 70% humidity and 12 h dark/12 h natural light for 60 days, the corms and roots of the banana seedlings were cut to observe the infection degree of Foc TR4 using a laser confocal microscope (Axio Scope A1, Carl ZEISS, Germany). The following growth and physiological parameters were measured: plant height, stem diameter, and plant biomass (fresh and dry weight). Additionally, the second fully expanded leaf from each plant was sampled for determining leaf area, leaf thickness, and relative chlorophyll content (using a SPAD meter). The disease index (DI) for *Fusarium* wilt was assessed based on a 0–6 scale: 0 = no symptoms; 1 = 1%–20%; 2 = 21%–40%; 3 = 41%–60%; 4 = 61%–80%; and 5 = 81%–100% of leaves affected or dead plant. The DI and biocontrol efficacy were calculated according to Chen et al. (16).

## Effect of strain SCA4-21 on the microbial community structure of banana rhizosphere soil

For each treatment, rhizosphere soil was collected from multiple plants by carefully uprooting them, vigorously shaking to remove loosely adhered bulk soil, and then brushing the firmly attached soil with sterile brushes. All rhizosphere soil from a given treatment was pooled, homogenized, and subdivided into three aliquots (biological replicates) for storage at −80°C. Total genomic DNA was extracted from each sample using the VAMNE Stool/Soil DNA Extraction Kit (Vazyme, China). The concentration and quality of the DNA were assessed using a Synergy HTX microplate reader (GeneCompany

Limited, Hong Kong, China) and 1.8% agarose gel electrophoresis, respectively. The bacterial 16S rRNA gene V3–V4 regions were amplified with primers 341F (5′- ACTCCTAC GGGAGGCAGCA-3′) and 806R (5′- GGACTACHVGGGTWTCTAAT-3′), while the fungal ITS1 region was amplified using primers ITS1F (5′-CTTGGTCATTTAGAGGAAGTAA-3′) and ITS2 (5′-GCTGCGTTCTTCATCGATGC-3′). The amplicons were purified, quantified, normalized, and constructed into sequencing libraries following quality control. Finally, the libraries were subjected to paired-end sequencing (2 × 250 bp) on an Illumina NovaSeq 6000 platform (Beijing Baimaike Biotechnology Co., Ltd., China).

The resulting raw sequencing data were processed on BMKCloud (https://www.bio-cloud.net). Briefly, low-quality sequences ($Q$-score < 20) and adapters were removed using Trimmomatic (version 0.33) (64) and Cutadapt (version 1.9.1) (65). Quality-filtered reads were denoised, merged, and chimera-checked using DADA2 in QIIME2 (version 2020.6) to generate amplicon sequence variants (66, 67). Taxonomy identification of ASVs was performed using the Naive Bayes classifier with the SILVA database (version 138) for bacteria and the UNITE database (version 8.0) for fungi at a 70% confidence threshold (68). Alpha diversity (Shannon, Simpson, Chao1, and ACE) and beta diversity (based on Binary-Jaccard distance for NMDS and Bray-Curtis for PCA) were calculated in QIIME2 (67). The Wilcoxon test and ANOSIM were used for significance testing of alpha and beta diversity at ASV level, respectively. A clustered heatmap of the top 20 most abundant genera was generated using the pheatmap package in R (69). Genus abundance was $Z$-score normalized (by row) and then clustered for both rows and columns using Euclidean distance and complete linkage. Differentially abundant genera were identified using ANCOM-BC (70). Significance levels in figures are indicated as *$P$ < 0.05, **$P$ < 0.01, ***$P$ < 0.001, and ns (not significant).

Bacterial functions were predicted using Tax4Fun2 (71), and KEGG pathway level 2 abundance was analyzed. Differential abundance of the top 20 KEGG pathways (level 2) across sample groups was tested using Welch's two-sample $t$-tests (a type of pairwise comparison) as implemented in the STAMP software package, with a significance threshold of $P$ < 0.05. Bacterial co-occurrence networks were constructed to compare the microbial community structure under different treatments. This was achieved by calculating pairwise Spearman correlations between ASVs and using relationships with $|r|$ > 0.1 and $P$ < 0.05 to build networks in the Molecular Ecological Network Analysis pipeline. The global topological properties (e.g., connectivity and modularity) of the resulting networks were then calculated and compared. Additionally, putative keystone taxa were identified based on network topology (72).

## Statistical analysis

Statistical analysis was performed with the SPSS Version 22.0 software (SPSS Inc., Chicago, IL, USA). Significant differences between means were determined by Duncan's multiple range test at $P$ < 0.05. All data from three biological replicates of each experiment were obtained as means ± the standard error.

## ACKNOWLEDGMENTS

This study was supported by the project of the Open Funds of State Key Laboratory of Tropical Crop Breeding (SKLTCBKF202501), National Key Laboratory for Tropical Crop Breeding (NKLTCB202306), the National Natural Science Foundation of China (U22A20487 and 322MS126), the Natural Science Foundation of Hainan (322QN417 and 322MS126), Chinese Academy of Tropical Agricultural Sciences for Science and Technology Innovation Team of National Tropical Agricultural Science Center (CAT-ASCXTD202309 and CATASCXTD202312), Central Public-interest Scientific Institution Basal Research Fund (1630052023011), and the China Agriculture Research System (CARS-31).

## AUTHOR AFFILIATION

[1]Institute of Tropical Bioscience and Biotechnology, National Key Laboratory of Biological Breeding of Tropical Crops, Chinese Academy of Tropical Agricultural Sciences, Haikou, China

## AUTHOR ORCIDs

Wei Wang http://orcid.org/0000-0003-1436-1395
Dengfeng Qi http://orcid.org/0000-0003-4711-9992
Jianghui Xie http://orcid.org/0000-0001-5298-3107

## AUTHOR CONTRIBUTIONS

Qiao Liu, Investigation, Validation, Writing – original draft | Liangping Zou, Formal analysis, Visualization, Writing – review and editing, Writing – original draft | Yufeng Chen, Methodology, Validation | Junting Feng, Conceptualization, Visualization | Yongzan Wei, Data curation, Formal analysis | Miaoyi Zhang, Formal analysis, Resources | Kai Li, Data curation, Formal analysis | Yankun Zhao, Investigation, Validation | Dengbo Zhou, Resources, Validation | Wei Wang, Funding acquisition, Supervision, Writing – review and editing | Dengfeng Qi, Funding acquisition, Methodology, Project administration, Supervision, Writing – original draft, Writing – review and editing | Jianghui Xie, Funding acquisition, Methodology, Project administration, Supervision, Writing – original draft, Writing – review and editing

## DATA AVAILABILITY

The raw sequence data reported in this paper have been deposited in the NCBI Sequence Read Archive (SRA) under accession numbers SRR26906767–SRR26906784.

## ADDITIONAL FILES

The following material is available online.

### Supplemental Material

**Supplemental material (Spectrum02968-24-S0001.docx).** Fig. S1; Tables S1 to S5.

### Open Peer Review

**PEER REVIEW HISTORY (review-history.pdf).** An accounting of the reviewer comments and feedback.

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
