## [Reviewer comments · Microbiology Spectrum]

Microbiology Spectrum

Biocontrol efficiency and mechanism of novel *Streptomyces luomodiensis* SCA4-21 against banana *Fusarium* wilt

Qiao Liu, Liangping Zou, Yufeng Chen, Junting Feng, Yongzan Wei, Miaoyi Zhang, Kai Li, Yankun Zhao, dengbo zhou, Wei Wang, Dengfeng Qi, and Jianghui Xie

Corresponding Author(s): Qiao Liu, Institute of Tropical Bioscience and Biotechnology

Review Timeline:

Submission Date:	March 20, 2025
Editorial Decision:	August 14, 2025
Revision Received:	October 20, 2025
Accepted:	October 29, 2025

Editor: Frédérique Reverchon

Reviewer(s): The reviewers have opted to remain anonymous.

Transaction Report:

DOI: <https://doi.org/10.1128/spectrum.02968-24>

Re: Spectrum02968-24 (Biocontrol efficiency and mechanism of novel *Streptomyces luomodensis* SCA4-21 against banana *Fusarium* wilt)

Dear Dr. Qiao Liu:

Thank you for the privilege of reviewing your work. Below you will find my comments, instructions from the Spectrum editorial office, and the reviewer comments.

I have received the comments made by one reviewer and have assessed the manuscript myself.

Please find below both assessments:

Comments by the Editor

Methods should be described in more detail, especially the VOC experiment, sequencing and bioinformatics analyses. ANOVA is not suited for compositional data so you may want to explore other alternative statistics for your metabarcoding data (ANCOM for example). A proper control in the *in vivo* experiment is lacking ("healthy plants" should have been immersed as well for 30 min).

The rationale behind some experiments should be better introduced. For example, why testing the extracts against pathogens that are not soil-borne? Why not testing the extracts in the *in vivo* experiment?

Minor comments:

L53: Link the first sentence and the second. The switch to the description of the pathogen is rather abrupt. Maybe some data regarding the incidence of the pathogen, and its worldwide relevance?

L54: "Foc TR4 is susceptible to almost all banana cultivars". The other way around?

L61: alternative to what?

L65-75: please focus on soil-borne pathogens to be consistent with your previous sentences about rhizosphere *Streptomyces* for the control of soil-borne pathogens. Not all the cited studies refer to rhizospheric strains.

L71: "found", not "founded"

L77: *in vitro* in italics

Rather than a description of the results in the Introduction, I would rather prefer to see some background regarding the potential of rhizosphere strains as biocontrol agents of FOC TR4 in banana and a more in depth description of studies cited as 15, 16 and 17 (antagonistic activity of *Streptomyces* strains against FOC TR4). This would help emphasise the relevance of the present study. Please provide a clear objective for this study instead of enlisting the experiments you performed in the Introduction.

L500: colonial instead of clonal?

L504: did you consider using a positive control for the antifungal activity, for example a commercial fungicide used in the treatment of FOC TR4?

L514: The statement that the extracts were tested against 8 pathogenic fungi is misleading, since some of these fungi are the same species (albeit different strains). What was the purpose of testing the extracts against these fungi? Some of them are not soil-borne, so this should be justified. Are they all affecting banana?

Replicates for section 2.4?

L533: "incubating", not "incubate".

L542: how many PDA plates?

L567: were inoculated

Please describe the methods for capturing the VOCs in more details: type of SPME fibers? Parameters for the GC-MS analysis? Compound database? I understand it was described in another paper but details should be added here as well.

Please ensure the titles for subsections of the Methods are homogenized (some sections are numbered, others are not).

The choice of inoculating the strain and not applying compounds (either extracts or VOCs) for biocontrol should be justified. The whole manuscript is aimed at determining the metabolites produced by this *Streptomyces* strain so the focus of the *in vivo* biocontrol experiment seems rather odd.

L586: control for the *in vivo* experiment should have been immersed in dH₂O for 30 min to consider the effect of the bathing stress.

L608: how was DNA extraction performed? How were the sequencing library prepared? Which primers were used? What was the sequencing depth?

L614: Why OTUs and not ASVs?

L619: OTUs, not OUT

L627: ANOVA is not suited for compositional data.

What was the purpose of the bacterial network analysis? Networks based on correlations of taxa relative abundance may not reflect real ecological interactions. Please provide parameters for the network construction.

Fig. 2C: could images with the same scale could be provided for better comparison?

L105: "effect" instead of "manner"?

Fig. 6A and 6B are redundant? I understand they present the same data. Presenting histograms at the genus level makes them hard to interpret, maybe at the class level? Differentially enriched genera could be emphasised in another way, through DeSeq for example. Same comment for the fungal community.

L206: there was quite a lot of dispersion in my opinion. What makes you say that "Principal component analysis (PCA) revealed a tendency for different samples of the same treatment to cluster together"? Same comment for the fungal community. Please do not enlist all relative abundance percentages as it makes the reading quite tedious. Emphasise your main results and differences between treatments.

L319: *in vitro* in italics

L339-342: why not focus on antifungal activities exhibited by these VOCs? You should acknowledge that the VOC profile of *Streptomyces* is likely to change when in contact with the pathogen or the soil native microbiota.

L374-379: not surprising since correlation networks are based on your relative abundance results. Yet these are correlations and do not mean that there is a regulatory effect of your beneficial strain on these taxa, please rephrase. These positive correlations may be due to other factors.

The predicted functions (Tax4fun) are not discussed. Why ?

Revision Guidelines

Sincerely,
Frédérique Reverchon
Editor
Microbiology Spectrum

Reviewer #3 (Comments for the Author):

This manuscript presents a well-designed and comprehensive study investigating the antifungal efficacy and biocontrol mechanisms of *Streptomyces luomodensis* SCA4-21 against *Fusarium oxysporum* f. sp. *cubense* TR4 (Foc TR4). The

experimental framework is sound, and the integration of physiological, microbiological, metabolomic, and molecular data greatly strengthens the validity of the conclusions.

Strengths

- The identification and functional characterization of a new *Streptomyces* species with potent antifungal activity is timely and highly relevant to managing banana *Fusarium* wilt.
- The combined use of GC-MS, high-throughput sequencing, and plant growth experiments provides a holistic view of the biocontrol mechanism.
- The beneficial effects on the rhizosphere microbiome and plant health are clearly demonstrated and supported by both qualitative and quantitative data.

Improvements:

- **Statistical Rigor and Transparency:** Clarify assumptions for ANOVA tests (e.g., normality, homogeneity of variance).
- Report exact p-values or significance indicators (ns, *, **) in all figure panels.
- **Clarify replicates:** Define biological vs. technical replicates explicitly.
- Indicate post-hoc test results in figure legends (e.g., Duncan's test letters).

Reference and Citation Consistency (Please review and reconcile all citations to ensure consistency).

- Several references are cited in-text but missing from the reference list, including: Ofek et al. (2012), Kuffner et al. (2010), Grönemeyer et al. (2011), etc.
- Conversely, some references are listed but not cited (e.g., Weller 1988, Bérdy 2005).

Figures and Tables

- Add or clarify scale bars and magnifications in SEM/TEM images (Fig. 2C-E).
- Standardize color coding and labeling across all graphs (T1, T2, T3).
- Ensure all axes and units are clearly labeled (e.g., g, cm, %, mg/mL).
- Legends should be self-contained-include treatment details and significance explanations.

Language and Grammar

- Minor grammatical corrections are needed throughout (e.g., "founded" → "found"; "significantly reduced the abundances" → "significantly reduced the abundance").
- Consider revising long and dense sentences for better readability.

Discussion

- Expand on potential applications of strain SCA4-21 in commercial biocontrol products.
- Briefly discuss limitations or next steps, such as testing in field conditions or formulation development.

Formatting and Structure

- Reorganize long paragraphs into subheadings within the "Discussion" for better flow (e.g., "Volatile Metabolites", "Microbial Community Changes").

**Biocontrol efficiency and mechanism of novel *Streptomyces luomodiensis* SCA4-21**
**against banana *Fusarium* wilt**

**Qiao Liu[#], Liangping Zou[#], Yufeng Chen, Juntao Feng, Yongzan Wei, Miaoyi**
**Zhang, Kai Li, Yankun Zhao, Dengbo Zhou, Wei Wang^{*}, Dengfeng Qi^{*}, Jianghui**
**Xie^{*}**

Institute of Tropical Bioscience and Biotechnology, National Key Laboratory of Biological
Breeding of Tropical Crops, Chinese Academy of Tropical Agricultural Sciences, Haikou, 571101,
China

**Correspondence:** Dr. Wei Wang (wangweisys@ahau.edu.cn), Dengfeng Qi
(qidengfeng@itbb.org.cn); Dr. Jianghui Xie (xiejianhui@itbb.org.cn)

**Keywords:** *Streptomyces luomodiensis*, biocontrol, banana *Fusarium* wilt, soil
**microbiome, extracts, antifungal activity**

**ABSTRACT**

The soil-borne fungi *Fusarium oxysporum* f. sp. *cubense* tropical race 4 (Foc TR4)
causes banana *Fusarium* wilt, which seriously threatens global banana production.
Biocontrol has been considered viable alternative method to manage banana *Fusarium*
wilt. In previous study, **we found** that novel *S. luomodiensis* SCA4-21 exhibited
antifungal activity. Here, we revealed that the extracts of strain SCA4-21 significantly
inhibited the growth of Foc TR4 hyphae and spore germination, severely disrupting the
ultrastructure of Foc TR4 hyphal cells and spore morphology. These extracts also
exhibited broad-spectrum antifungal activity against eight other phytopathogenic fungi.
Furthermore, we demonstrated that strain SCA4-21 produce 32 volatile organic
compounds, including five antifungal compounds. **In a pot experiment**, we discovered
that the inoculation of strain SCA4-21 significantly inhibited the infection of Foc TR4

in banana seedling corms, achieving a biocontrol efficiency of 59.3%, and promoted
the growth of banana seedlings. Additionally, this inoculation significantly enhanced
the abundances of beneficial bacterial genera *Streptomyces*, *Bacillus*, *Sphingomonas*,
and *Massilia*, as well as fungal genera *Mortierella*, *Purpureocillium*, *Gibellulopsis*, and
*Xenomyrothecium*, while significantly reduced the abundances of pathogenic bacteria
genus *Pantoea* and fungal genus *Fusarium*, in the banana rhizosphere soil. Moreover,
the inoculation of strain SCA4-21 significantly increased the enrichment of pathways
such as carbohydrate metabolism, amino acid metabolism, and metabolism of
terpenoids and polyketides. Therefore, we postulated that strain SCA4-21 may
synergistically combat banana *Fusarium* wilt by producing antifungal compounds and
enriching beneficial bacteria and fungi. Our findings present a promising biocontrol
agent for the management of banana *Fusarium* wilt.

**IMPORTANCE**

Banana (*Musa* spp.) is one of the most popular fruit crops and the fourth largest food
crop in developing countries within tropical and subtropical regions. However, the
emergence and rapid spread of strain Foc TR4 seriously hinder the development of
banana industry. Currently, there is no effective control measure available. Biological
control holds potential due to its safety and effectiveness. Here, we found that the
extracts of *S. luomodiensis* SCA4-21 exhibited significant inhibitory effects on the
hyphal growth and spore germination of Foc TR4, as well as severe destructive effects
on its cell morphology and ultrastructure, and broad-spectrum antifungal activity. We
also discovered that the inoculation of strain SCA4-21 significantly inhibited the
infection of Foc TR4 in banana seedling corms, reduced the incidence index of banana
*Fusarium* wilt, promoted the growth of banana seedlings, and enhances beneficial
microbes and metabolic pathways, suggesting that strain SCA4-21 is a promising

biocontrol age
INTRODUCTION

Banana is a major source of starch in tropical and subtropical developing countries

(1). Banana *Fusarium* wilt caused by Foc TR4 is a devastating soil-borne fungal

disease. Foc TR4 is susceptible to almost all banana cultivars (2). In the absence of

living host tissues, Foc TR4 can survive in soil as chlamydospores for over 20 years.

Once the environmental conditions become favorable, the pathogen invades the root

system of the banana plant and rapidly spreads. Its proliferating mycelium and

secretions obstruct the vascular system, inducing water stress that leads to wilting and

eventual mortality of the plant (3).

The utilization of biocontrol agents for managing banana *Fusarium* wilt is regarded as

a viable alternative approach owing to its efficacy and eco-friendliness (4).

Rhizosphere *Streptomyces* spp. are ideal candidates for biocontrol agents against soil-

borne diseases due to their ability to actively colonize plant root systems and survive

for a long time even under extreme condition such as low nutrition and water

availability in the form of spores (5-8). They have been studied extensively as

potential biocontrol agents against fungal phytopathogens such as *Magnaporthe*

*oryzae*, *Colletotrichum fragariae*, and *Colletotrichum gloeosporioides* (9-11). They

protect crops from diseases by synthesizing diverse bioactive secondary metabolites,

competing with pathogens for nutrients by producing substances like siderophores, or

producing a large number of extracellular enzymes related to fungal cell wall

degradation (6,12-14). We also founded that *Streptomyces* spp. has potential

application prospects in the treatment of banana *Fusarium* wilt caused by Foc TR4

(15-17).

In our previous study, strain SCA4-21 was isolated from wheat rhizosphere soil in a
dry-hot valley and identified as a new species of *Streptomyces*, named after
*Streptomyces luomodiensis* sp. nov.. *S. luomodiensis* SCA4-21 exhibited excellent
antagonistic activity against nine phytopathogenic fungi including Foc TR4 in vitro
assay (18). In present study, the extracts of stain SCA4-21 were extracted and purified,
and the half maximal effective concentration (EC₅₀) of the extracts on Foc TR4 was
determined. The extracts were also tested for their broad-spectrum antifungal activity
against other eight plant pathogenic fungi. Additionally, the impact of the extracts on
mycelial growth, spore germination, hyphal and spore morphology, as well as the
ultrastructure of Foc TR4 was observed. Furthermore, volatile organic compounds of
strain SCA4-21 were identified using gas chromatography-mass spectrometry (GC-
MS). In addition, the biocontrol potential of strain SCA4-21 against banana *Fusarium*
wilt caused by Foc TR4 and its effects on the growth of banana seedlings was evaluated.
The effect of stain SCA4-21 on soil microbial communities was also analyzed through
high-throughput sequencing. These findings highlight the excellent potential of strain
SCA4-21 as a source of biocontrol agents.

**Result**

**Extraction and purification of strain SCA4-21 extracts**

The ethanol extracts of strain SCA4-21 fermentation broth were concentrated and
adsorbed onto macroporous resins, then eluted with methanol gradient solutions to
separate the antifungal extracts. The results showed that the extracts eluted with 100%
methanol exhibited the highest mycelial inhibition rate (71.57±0.08) against Foc TR4,
followed by the extracts eluted with 70% (31.53±0.15), 60% (17.67±0.25) and 50%

(9.39±0.17) methanol, respectively (Fig. 1A). In fact, as the concentration of methanol
increased, the antifungal activity of the extracts significantly enhanced. The extracts
with eluted with 100% methanol was selected for next study due to its strong antifungal
activity.

**Measuring the EC₅₀ of strain SCA4-21 extracts on the mycelial growth of Foc**

**TR4**

The half maximal effective concentration (EC₅₀) of strain SCA4-21 extracts on mycelial
growth of Foc TR4 was determined. Mycelial inhibition showed a dose-dependent
manner. After exposing Foc TR4 to the extracts at 28°C for 7 days, significant growth
inhibition of the pathogen was observed in all concentration groups (0.78, 1.56, 3.13,
6.25, 12.5, 25, 50, 100, and 200 mg l⁻¹). The inhibition of mycelial growth was as
follows: 13.25±0.12, 14.45±0.11, 17.27±0.13, 19.88±0.13, 24.50±0.06, 32.73±0.08,
40.16 ±0.05, 49.80 ±0.08, 61.65 ±0.09, respectively. EC₅₀ value is calculated as
37.88±1.58 µg mL⁻¹ using toxicity regression equation (Fig. 1B).

**Strain SCA4-21 extracts exhibiting broad-spectrum antifungal activity**

Strain SCA4-21 extracts showed effective antifungal activity against the tested strains
of fungi. Compared to the control treated with 10% DMSO, the fungal hyphae treated
with the extracts displayed significant inhibition. The inhibition rates for *C.*
*gloeosporioides* (Penzig) Penzig et Saccardo, *C. gloeosporioides*, *C. gloeosporioides*,
*F. graminearum* Sehw, *C. fragariae*, *C. gloeosporioides* (Penz) Saec, *F.*
*oxysporum f.sp cucumerinum*, *C. acutatum* were 68.37%, 53.69%, 50.23%, 46.52%,
45.84%, 43.23%, 37.92 and 25.44, respectively (Fig. 1C)

**Strain SCA4-21 extracts significantly inhibiting spore germination of Foc TR4**

The germination of spores was significantly inhibited after treated with extracts from
strain SCA4-21(Fig. 2A). The inhibitory efficiency showed a positive correlation with

the concentration of the extracts used in the treatment. Compared to the control group,
the spore germination rates of Foc TR4 were 65%, 44% and 20% after treatment with
$1 \times EC_{50}$, $2 \times EC_{50}$, $4 \times EC_{50}$, respectively. The inhibition efficiencies were calculated as
follows: 15.6%, 42.9%, and 74.0%. Furthermore, almost **complete inhibition** of spore
germination was observed when an extract concentration equivalent to $8 \times EC_{50}$ was
used (Fig. 2B).

**Strain SCA4-21 extracts severely destroying morphology and ultrastructure of** 129 **Foc TR4**

SEM observation revealed that the control mycelium of Foc TR4 treated with 10%
DMSO exhibited a smooth and intact morphology, while mycelium of Foc TR4 treated
with the $4 \times EC_{50}$ extracts displayed notable alterations such as swelling and fusion (Fig.
2C). Simultaneously, the treated spores also exhibited a disrupted and concave
morphology, whereas the control spores displayed a regular and planar appearance (Fig.
2D).

TEM analysis showed that the ultrastructure of Foc TR4 hyphae with the $4 \times EC_{50}$
extracts were **also seriously damaged**. Their organelles such as mitochondria and
vacuoles disintegrated. The cytoplasm shrinks, the cell membrane becomes indistinct
and the nucleus disappears. In contrast, the cell membrane and wall of the control group
were clear and complete, Cell vacuoles are filled and mitochondrial cristae are clearly
distinguishable (Fig. 2E).

**Identification of the volatile organic compounds of strain SCA4-21**

GC-MS analysis was conducted to identify the volatile organic compounds of strain
SCA4-21 that are likely responsible for its antifungal activities. by comparing the mass
spectra of treatments with those of controls, 32 secondary metabolites of *S.*
*luomodensis* SCA4-21 were determined, including Oxetane, 2,3,4-trimethyl-, (2,3

à,4á)-, Butanoic acid, methyl ester (Methyl butyrate), Ethene, fluoro-, Methyl
isovalerate, Methyl tiglate, Pentanoic acid, 3-methyl-, methyl ester, Pentanoic acid, 4-
methyl-, methyl ester, Hexanoic acid, methyl ester (methyl hexanoate), Thiopivalic acid,
6-Methyl-2-heptanone, 5-Methyl-2-heptanone, (E)-2-Hexenoic acid, methyl ester
(Methyl (E)-2-hexenoate) , Furyl hydroxymethyl ketone, 3-Octanone Hexanoic acid,
5-methyl-, methyl ester, Heptanoic acid, methyl ester, (Fig. 3, Supplementary Fig. 1A-
F and Table S1)

**Strain SCA4-21 significantly reducing the incidence of banana *Fusarium* wilt** 155 **disease**

The pot experiment revealed that Strain SCA4-21 significantly reduced the incidence
of banana wilt disease caused by Foc TR4 (Fig. 1A-B). Specifically, no evident
chlorotic leaves were observed in negative control group (T1) where banana plantlets
treated with sterilized SLM only. While, in positive control group (T2), the leaves of
banana plantlets treated with sterilized SLM and GFP-Foc TR4 exhibited **seriously**
chlorotic **symptom**, resulting in a disease index of 57.5%. In contrast, only a few old
leaves of banana seedlings treated with GFP-Foc TR4+SCA4-21 (T3) showed yellow
symptoms, leading to a significantly lower disease index of 23.4% compared to positive
controls (Fig. 4A). The biocontrol efficiency of strain SCA4-21 against banana
*Fusarium* wilt was 59.3% (Fig. 4B).

Similarly, the corm sections of the T1, T2, and T3 groups displayed white, dark
brown, and light brown colors respectively. Laser confocal microscope analysis
further revealed that only a small amount of mycelium was observed in the outer cells
of the T3 bulbs, unlike the corms in the T2 group where vascular cells were filled with
Foc TR4 hyphae, while the corms in the T1 group were not infected by Foc TR4 (Fig.
4A).

**Strain SCA4-21 promoting significantly the growth of banana seedlings**

The pot experiment also indicated that strain SCA4-21 significantly promoted the
growth of banana seedlings (Fig. 4C-H). The T3 group, treated with SCA4-21+Foc TR4,
exhibited significant increases in plant height (62.9%) (Fig. 4C), stem diameter (25.2%)
(Fig. 4D), chlorophyll content (24.3%) (Fig. 4E), leaf area (43.9%) (Fig. 4F), vane
thickness (14.9%) (Fig. 4G), fresh weight (115.5%) and dry weight (59.3) of banana
seedlings 60 days after planting when compared to the T2 group treated with SLM+Foc
TR4 (Fig. 4H).

Compared to the T1 group treated with SLM, the T3 group also showed significant
increases in plant height (23.3%), fresh weight (25.6%), and dry weight (9.6%).
Additionally, stem diameter, chlorophyll content, leaf area, and leaf thickness increased
by 5.9%, 9.1%, 8.9%, and 2.8% respectively; however, there was **no statistically**
**significant difference** between these four indicators in the two groups.

**Strain SCA4-21 shaping structure of banana rhizosphere bacterial community**

720,187 pairs of raw reads were obtained from 9 soil samples by sequencing the V3+V4
regions of bacterial 16S rRNA. These bacterial raw reads were submitted to NCBI's
Sequence Read Archive database (accession number: SRR26906769-74,
SRR26906783-4). Subsequently, after filtering, cutting and splicing, a total of 718,594
pairs of clean reads were obtained (Table 1). After the chimera was removed, the
obtained 482,135 non-chimeric clean reads were clustered into 11210 distinct bacterial
Operational Taxonomic Units (OTUs) with 97% similarity. Among them, 1732, 1899
and 1862 OTUs were derived **from T1** group soil samples treated with sterilized SLM.
while 1946, 1863 and 1851 OTUs were derived from T2 group soil samples treated
with sterilized SLM and GFP-Foc TR4. Similarly, 1602, 1852 and 1647 OTUs were
derived from T3 group soil samples treated with Foc TR4+SCA4-21(Fig. 5A). Venn

analysis revealed that T1, T2, and T3 groups had 3201,3408 and 3145 unique bacteria
OTUs respectively (Fig. 5B).

Analysis of bacterial alpha diversity showed that the diversity indices Shannon and
Simpson in T1 group soils were higher than those in T2 and T3 group soils (Fig. 5C-
D). While, T2 group soils showed higher the richness indices ACE and Chao1,
followed by the T1 and T3 group soils (Fig. 5E-F). However, there were no
statistically significant differences observed in the bacterial the diversity indices
Shannon and Simpson, as well as the richness indices ACE and Chao1, among the
three treatments (Fig. 5C-F and Supplementary Table S2).

Principal component analysis (PCA) revealed a tendency for different samples of the
same treatment to cluster together, indicating a relatively high similarity in bacterial
community composition among samples within the same group. PCA results also
showed that PC1 explained 66.89% of the total variation, PC2 explained 17.98% of
the total variation, suggesting good separation in bacterial community composition
among the treatment groups (Fig. 5G). Nonmetric multidimensional scaling (NMDS)
analysis result clearly showed significant variations in bacterial community
composition across the different treatments (ANOSIM, $R = 0.909$, $P=0.003$), and low
stress value (stress=0.0017<0.05) indicating that the analysis results have excellent
representativeness (Fig. 5H).

Taxonomic annotations showed that a total of 11,210 bacterial OTUs were classified
into 2 kingdoms, 40 phyla, 99 classes, 265 orders, 515 families, 971 genera and 1142
species (Table 2). The top 20 bacterial genera in terms of relative abundance were
displayed using bar chart (Fig. 6A) and heatmap (Supplementary Fig. S2A), including
unclassified-*Xanthobacteraceae* (5.32%, 4.97% and 2.29%, respectively), *Bacillus*
(0.30%, 0, and 9.55% respectively) unclassified-*Acidobacteriales* (5.49%, 3.54%, and

2.19% respectively), unclassified-*Bacteria* (3.17%, 3.37%, and 1.77% respectively),
unclassified-*Cyanobacteriales* (1.06%, 5.16%, and 0.32% respectively), *Burkholderia-*
*Caballeronia-Paraburkholderia* (1.81%, 0.91%, and 3.75% respectively), unclassified-
*Elsterales* (3.37%, 2.18%, and 0.85% respectively), *Haliangium* (2.53%, 2.43%, and
1.34% respectively), *Pantoea* (1.09%, 4.74%, and 0.05% respectively), *Bryobacter*
(3.22%, 1.38%, and 0.96% respectively), unclassified-*Gemmatimonadaceae* (1.94%,
2.30%, and 1.31% respectively), *Sphingomonas* (0.91%, 0.82%, and 2.95%
respectively), *Cupriavidus* (0.62%, 0.48%, and 3.37% respectively), unclassified-
*Comamonadaceae* (1.18%, 2.12%, and 1.14% respectively), unclassified-
*SBR1031*(2.49%, 1.52%, and 0.25% respectively), unclassified-*Alphaproteobacteria*
(1.86%, 1.73%, and 0.63% respectively), *Massilia* (0.18%, 0.05%, and 3.98%
respectively), *Pseudomonas* (0.33, 0.12%, and 3.71% respectively), uncultured-
*Acidobacteria-bacterium* (1.58%, 1.74%, and 0.59% respectively),
*Streptomyces*(0.48%, 0.31%, and 2.30% respectively) (Supplementary Table S3).

ANOVA method was used to analyze the differences among the top 20 bacterial genera
in three treatments (Fig. 6B). Compared to T1 and T2 groups soils, the soil samples of
the T3 group treated with Foc TR4+SCA4-21 exhibited significantly higher abundances
of *Streptomyces*, *Bacillus*, *Sphingomonas*, and *Massilia*. Conversely, the abundances of
*uncultured-Acidobacteria-bacterium*, *unclassified-Elsterales*, *unclassified-*
*Alphaproteobacteria*, *unclassified-Acidobacteriales* and *Pantoea* is significantly lower
in T3 group soils than in T1 and T2 group soils.

Bacterial functional profiles based on the Kyoto Encyclopedia of Genes and Genomes
(KEGG) was predicted via Tax4Fun (Fig. 7A-C). Compared to groups T1 and T2, T3
group exhibited significant enrichment in pathways related to carbohydrate metabolism,

amino acid metabolism, lipid metabolism, metabolism of terpenoids and polyketides,
and biosynthesis of other secondary metabolites (Fig. 7A-B). Conversely, when
comparing T1 and T3 groups, pathways such as nucleotide metabolism and translation
were significantly increased in group T2 (Fig. 7A, 7C). Additionally, pathways such as
folding, sorting and degradation, and replication and repair were significantly increased
in both T2 and T3 groups compared to group T1 (Fig. 7B-C).

In order to explore the competitive or cooperative relationships between *Streptomyces*
and other bacterial genera in soil samples treated differently, Spearman's rank
correlation coefficients were calculated and co-occurrence network of top 20 bacterial
genera by abundance was constructed. The results revealed that the relative abundance
of *streptomyces* was positive with *Sphingomonas*, *Burkholderia-Caballeronia*-
*Paraburkholderia*, *Massilia*, *Pseudomonas*, *Bacillus*, and *Cupriavidus*, while it was
negatively correlated with *Pantoea* and unclassified *Cyanobacteriales* (Fig. 7D).

**Strain SCA4-21 monitoring structure of banana rhizosphere fungal community**

Fungal ITS analysis of 9 soil samples yielded 720,798 pairs of raw reads. These fungal
raw reads were submitted to NCBI's Sequence Read Archive database (accession
number: SRR26906767, SRR26906775-82). Following splicing, chimaera removal,
and denoising procedures, we obtained a set of high-quality clean reads comprising
690,176 pairs (Table 3). These authentic biological sequences were subsequently
clustered into 2454 fungal OTUs at a similarity threshold of 97%. Specifically, the three
soil samples from T1 group contributed 309, 411 and 590 OTUs, respectively. The three
soil samples from T2 group comprised 316, 446 and 429 OTUs, respectively. The three
soil samples from T3 group generated 413, 341 and 480 OTUs, respectively (Fig. 8A).
Venn analysis demonstrated that T1, T2, and T3 groups had 677,615 and 719 unique
fungal OTUs respectively (Fig. 8B).

Analysis of fungal alpha diversity indicated that the diversity indices Shannon and
Simpson in T3 group soils were higher than those in T1 and T2 group soils (Fig. 8C-
D), while the richness indices Chao1 and ACE are slightly higher in T2 group soils than
in T1 and T3 group soils (Fig. 8E-F). However, no statistically significant differences
were observed in the fungal diversity indices Shannon and Simpson, as well as the
richness indices ACE and Chao1, among the three treatments (Fig. 8C-F and
Supplementary Table S4).

PCA results revealed that the fungal community composition in group T2 differs
significantly from groups T1 and T3, while the difference in fungal community
composition between groups T1 and T3 is relatively small (Fig. 8G). Furthermore, the
PCA results showed that the T2 group had a positive contribution to the total variation,
while the T1 and T3 groups had a negative contribution (Fig. 8G). The PCA analysis
also revealed that PC1 accounted for 37.46% of the total variation, and PC2 explained
20.32% of the total variation, indicating a clear separation in fungal community
composition among the treatment groups (Fig. 8G). The NMDS analysis revealed
significant differences in fungal community composition among the different
treatments (ANOSIM, $R = 0.481$, $P = 0.003$) with a low stress value ($\text{stress} = 0.0446 <$
0.05), suggesting the highly representative nature of the analysis results (Fig. 8H).

Taxonomic annotations showed that a total of 2454 fungal OTUs were identified into 1
kingdom, 11 phyla, 42 classes, 105 orders, 217 families, 400 genera and 537 species
(Table 3). The fungal genera in the top 20% relative abundance were illustrated using
292 bar charts (Fig. 9A) and heatmaps (Supplementary Fig. S2B). **These identified**
**dominant genera included *Fusarium* (6.65%, 37.29% and 16.92%, respectively),**
**unclassified- *Sordariomycetes* (7.34%, 21.36% and 6.98%, respectively), unclassified-**
***Basidiomycota* (15.13%, 3.68% and 12.06%, respectively), unclassified-*Ascomycota***

(7.56%, 6.78% and 3.89%, respectively), unclassified-*Agaricomycetes* (8.96%, 6.35%
and 1.58%, respectively), unclassified-Fungi (4.59%, 3.35% and 3.77%, respectively),
unclassified-*Dothideomycetes* (1.52%, 4.03% and 1.17%, respectively), *Gymnopilus*
(5.56%, 0.01% and 0, respectively), *Conocybe* (2.43%, 0.65% and 2.48%, respectively),
*Gibellulopsis* (0.39%, 0.3% and 2.51%, respectively), *Xenomyrothecium* (0.12%, 0.16%
and 2.74%, respectively), unclassified-*Thelephoraceae* (2.67%, 0 and 0, respectively),
*Purpureocillium* (0.34%, 0 and 2.28%, respectively), *Lycoperdon* (1.3%, 0 and 1.28%,
respectively) *Mortierella* (0.23%, 0.06% and 2.00%, respectively), *Cladosporium*
(0.52%, 0.33% and 1.41%, respectively), unclassified-*Glomeraceae* (0.86%, 0.68%
and 0.33%, respectively), *Rhizophagus* (1.11%, 0.54% and 0.16%, respectively),
*Arthrobotrys* (0.05%, 0.17% and 1.27%, respectively) and unclassified-*Glomeromycota*
(0.56%, 0.54% and 0.34%, respectively) (Supplementary Table S5).

ANOVA (Analysis of Variance) method was also used to analyze the differences
among the top 20 fungal genera in three treatments (Fig. 9B). Compared to Compared
to T1 and T2 groups soils, the T3 groups soil showed significantly higher
abundances of *Gibellulopsis* and *Xenomyrothecium*. In contrast, the abundances of
*Fusarium* significantly decreased in T3 groups soil than that in T2 groups soil.
Although the *Fusarium* abundance in T3 group soil was still higher than that in T1
group soil, there was no significant difference between them.

**Discussion**

*Streptomyces* spp. represents a significant source of microbial bioagents. The
production of antimicrobial bioactive substances undeniably constitutes a crucial
mechanism through which *Streptomyces* spp. exert their function as biocontrol agents.
Behera et al. demonstrated that *S. chilikensis* RC1830 exhibited in vitro antifungal
activity against *F. oxysporum* through the production of metabolites, leading to a

significant reduction (80.51%) in disease severity index of rice wilt disease in pot
experiments (19). Chen et al. also reported that *S. plicatus* B4-7 showed the ability to
synthesize the antifungal compound borrelidin, resulting in a significant reduction of
75% in crown rot disease incidence (20). Our previous studies have also shown that
some *Streptomyces* strains such as *Streptomyces* sp. CB-75 and *Streptomyces* sp.
YYS-7 significantly decreased the incidence of *Fusarium* wilt of banana by inhibiting
mycelium growth, spore germination and destroying cell ultrastructure of Foc TR4
through secondary metabolites (15, 21).

In this study, we also discovered that extracts of *S. luomodiensis* SCA4-21
significantly inhibited the growth of Foc TR4 hyphae (Fig. 1A-B) and spore
germination (Fig. 2A-B), severely disrupted the ultrastructure of Foc TR4 hyphal cells
and spore morphology (Fig. 2C-E). The extracts also showed broad-spectrum
antifungal activity against eight other phytopathogenic fungi (Fig. 1C). GC-MS
analysis revealed that strain SCA4-21 produces a total of 32 volatile organic
compounds, including Butanoic acid, methyl ester, Methyl isovalerate, 5-Methyl-2-
heptanone, 6-Methyl-2-heptanone, 3-Octanone, Hexanoic acid, 2-ethyl-, methyl ester,
and Benzoic acid, methyl ester, among others (Fig. 3 and Supplementary Table S5).

The previous studies have indicated that these compounds possess specific biological
activities. Khan et al. reported that Butanoic acid, methyl ester (Methyl butyrate), at a
concentration as low as 0.01 M, induced significant cytotoxicity in the human breast
cancer cell line MDA-MB-231 (22). Ayaz et al. discovered that Methyl isovalerate
exhibited strong nematicidal activity, achieving a mortality rate of 83% (23). Morita et
al. and Zhang et al. demonstrated that 5-Methyl-2-heptanone and 6-Methyl-2-
heptanone exhibit potent antifungal activities against *Penicillium italicum* and
*Alternaria solani*, respectively (24-25). Dotson et al. reported that the *Streptomyces*

volatile 3-octanone could modulate auxin/cytokinin levels, promoting growth in
*Arabidopsis thaliana* via the gene family *KISS ME DEADLY* (26). Our previous
findings also revealed that Hexanoic acid, 2-ethyl-, methyl ester (Methyl 2-
ethylhexanoate) , derived from *Streptomyces corchorusii* CG-G2, exhibits
significant antifungal activity against *Colletotrichum gloeosporioides* (27). Lima et al.
demonstrated that Benzoic acid, methyl ester (methyl benzoate) exhibits antifungal
activity against *Candida albicans* (28). These results suggest that the compounds
identified are responsible for the antifungal activity of *S. luomodensis* SCA4-21.
In this study, we also evaluated biocontrol efficiency of strain SCA4-21 against
banana *Fusarium* wilt. Our results demonstrated that strain SCA4-21 could suppress
mycelial infection of Foc TR4 in roots, resulting in a remarkable reduction by 59.3%
in disease incidence banana *Fusarium* wilt, and promote significantly the growth of
banana seedlings (Fig. 4A-H). The rhizosphere microbiome serves as the first line of
defense against soilborne pathogens (29). Therefore, it is essential to analyze the
effects of bioagent inoculation on the rhizosphere microbiome in the biocontrol of
soil-borne diseases. Our results of current high-throughput sequencing showed a
significant increase in the abundances of the genus *Streptomyces*, *Bacillus*,
*Sphingomonas*, and *Massilia* in the T3 group (treated with Foc TR4+SCA4-21),
compared to both the T1 group (treated with sterilized SLM) and the T2 group
(treated with sterilized SLM+ Foc TR4) (Fig. 6A-B). Moreover, T3 group also
exhibited **higher the abundances** of the genus *Pseudomonas*, *Burkholderia*-
*Caballeronia-Paraburkholderia* and *Cupriavidus* than the other two groups (Fig. 6A-
B). We also observed that the abundance of the bacterial genus *Pantoea* significantly
decreased in the T3 group compared to that in the T1 and T2 groups (Fig. 6A-B).
Therefore, we propose that the introduction of strain SCA4-21 may directly or in

directly attribute to the increased abundance of *Streptomyces*, *Bacillus*,
*Sphingomonas*, *Massilia*, *Burkholderia-Caballeronia-Paraburkholderia*, and
*Cupriavidus*, while simultaneously reducing the prevalence of *Pantoea* in the T3
group. Spearman network correlation analysis was conducted to identify associations
between *Streptomyces* and other bacterial genera, revealing a positive correlation
between *Streptomyces* and *Bacillus*, *Sphingomonas*, *Massilia*, *Burkholderia-*
*Caballeronia-Paraburkholderia*, and *Cupriavidus*, while showing a negative
correlation with *Pantoea* genus (Fig. 7D). This further confirms the regulatory effect
of strain SCA4-21 on these bacteria
The accumulated data showed that the genera *Bacillus*, *Sphingomona* and
*Pseudomonas* are important resources for biological control agents (30-34). Actually,
these beneficial bacteria are often recruited and enriched in plant roots by microbial
bioagents, playing an important role in defending against soil-borne diseases. Zhang
et al. reported that *Trichoderma asperellum* M45a triggered watermelon resistance to
*Fusarium* wilt by regulating rhizosphere microbes, including increasing the
abundance of the genera *pseudomonas* and *Sphingomonas* in soil (35). Yang et al.
found that *Streptomyces aureovorticillatus* HN6 could colonize the banana
rhizosphere soil and effectively inhibited banana *Fusarium* wilt disease by attracting
*Bacillus* and *Pseudomonas* to colonize the banana plant rhizosphere and decreasing
the relative abundances of *Fusarium* (36). Our previous studies also revealed that the
genera *Bacillus*, *Sphingomonas*, and *Pseudomonas* were significantly enriched in the
rhizosphere soil of banana plants free from *Fusarium* wilt compared to those infected
by the fungus (37). Additionally, the genus *Massilia* has been frequently documented
as a colonizer of both the rhizosphere and endorhizal environments (Ofek et al., 2012;
Kuffner et al., 2010). Some *Massilia* isolates have also been found to control plant

pathogens by secreting siderophore or extracellular enzymes (Hrynkiewicz et al.,
2010; Groñnemeier et al., 2011). The applications of the genera *Burkholderia*-
*Caballeronia-Paraburkholderia* and *Cupriavidus* in biocontrol remain undocumented;
however, they are frequently identified as dominant genera in various foods, such as
broad-bean paste and tea, suggesting their role as beneficial bacteria (36; Zeng et al.,
2020). However, *Pantoea* spp. are frequently recognized as plant pathogens (Wensing
et al., 2010; Dussault et al., 2017). Mwaheb et al. (2017) also demonstrated that the
inoculation of *Hirsutella minnesotensis* significantly reduced the abundance of the
genus *Pantoea* in nematode-infested soils.

Our findings also indicate that the introduction of Foc TR4 significantly enhanced the
abundance of the fungal genus *Fusarium* in the T2 group, while the inoculation of strain
SCA4-21 significantly decreased *Fusarium* abundance in the T3 group (Fig. 9A-B).
This suggests that strain SCA4-21 inhibits the proliferation of Foc TR4 in banana
rhizosphere soil. The severity of banana *Fusarium* wilt is accompanied by an increase
in *Fusarium* colonization in the field (37). The microbial bioagents for banana
*Fusarium* wilt often exhibit an inhibition of the proliferation of *Fusarium* in soil. Shen
et al. revealed that *Bacillus amyloliquefaciens* NJN-6 could more effectively control
*Fusarium* wilt disease by suppressing *Fusarium* growth in the field (38). We also
observed that the relative abundances of the fungal genera *Mortierella*, *Purpureocillium*,
*Gibellulopsis* and *Xenomyrothecium* in the T3 group soils were significantly higher
than those in the T1 and T2 group soils (Fig. 9A-B), Especially the abundance of
*Mortierella* is approximately 9 and 33 times higher in the T3 group than in the T1 and
T2 groups, respectively (Supplementary Table S5), suggesting strain SCA4-21 greatly
promoted the enrichment of these fungal genus in the banana rhizosphere soil.
*Mortierella* spp. are frequently recognized as beneficial fungi. Soman et al. found that

*Mortierella vinacea* exhibited antifungal and antibacterial activity (39). Li et al. (2014)
reported that relative abundance of *Mortierella* dropped significantly in the consecutive
peanut monoculturing fields. Ye et al. found that an increase in the relative abundance
of *Fusarium* leads to a decrease in the relative abundance of *Mortierella* (40). We
previously indicated that *Mortierella* (36.64%) was predominant fungal genus in the
*Fusarium* wilt disease-free soils (37). Hamid et al. demonstrated that *Mortierella* and
*Purpureocillium* are related to soil suppressiveness against soybean cyst nematode (41).
Additionally, the genus *Purpureocillium* is recognized as a promising biological control
agent, effectively parasitizing the eggs of plant-parasitic nematodes (42). Lopez et al.
found that the endophytic fungus *Purpureocillium lilacinum* enhances the growth of
cultivated cotton (*Gossypium hirsutum*) and inhibits cotton bollworm (*Helicoverpa zea*)
(43). Certain species within the genus *Gibellulopsis* are classified as pathogenic fungi,
such as *Gibellulopsis nigrescens*, which causes wilt in sugar beets (44). However,
numerous hypovirulent strains exist, including *Gibellulopsis nigrescens* Vn-1 and
*Gibellulopsis nigrescens* CVn-WHg, which are employed as biological control agents
against Verticillium wilt in sunflowers and cotton (45-46). Jin et al. also discovered that
crop rotation increased the relative abundances of *Gibellulopsis* in the cucumber
rhizosphere, indicating the beneficial role of the genus *Gibellulopsis* in healthy soils
(47). The genus *Xenomyrothecium* is classified within the group of saprophytic fungi,
which play a crucial role in the decomposition of plant biomass, highlighting their
ecological significance (48). Furthermore, Sundar and Arunachalam discovered that
*Xenomyrothecium tongaense* PTS8, a rare endophyte of *Polianthes tuberosa*, exhibits
significant antagonism against multidrug-resistant pathogens (49).
Therefore, we postulated that strain SCA4-21 may synergistically control banana
*Fusarium* wilt by producing antifungal compounds and enriching beneficial bacteria

and fungi. The functions of bacteria and fungi still needs to be further explored in the
decrease of *Fusarium* abundance in soil.

**5. Conclusion**

The extracts of *S. luomodensis* SCA4-21 significantly inhibited hyphae growth and
spore germination of Foc TR4, severely destroyed the ultrastructure of Foc TR4 hyphal
cells and spore morphology, and exhibited excellent broad-spectrum antifungal activity
against eight other phytopathogenic fungi. GC-MS analysis revealed that strain SCA4-
21 produces a total of 32 volatile organic compounds, including the anticancer
compound Butanoic acid methyl ester, the nematicidal compound Methyl isovalerate,
the growth-promoting compound 3-Octanone, and the antifungal compounds 5-Methyl-
2-heptanone, 6-Methyl-2-heptanone, Hexanoic acid methyl ester, and Benzoic acid
methyl ester. Pot experiment revealed that the inoculation of strain SCA4-21
significantly inhibited infection of Foc TR4 corms of banana seedlings, resulting in a
notable decrease in the disease index and an improvement in the growth of banana
seedlings. High-throughput sequencing indicated that the inoculation of strain SCA4-
21 significantly enhanced the population of beneficial microorganisms, including
bacterial genera like *Streptomyces*, *Bacillus*, *Sphingomonas*, and *Massilia*, and fungal
genera such as *Mortierella*, *Purpureocillium*, *Gibellulopsis*, and *Xenomyrothecium*,
while significantly reducing pathogens such as the bacterial genus *Pantoea* and the
fungal genus *Fusarium* in banana rhizosphere soil. Spearman bacterial correlation
network analysis revealed that the genus *Streptomyces* exhibited a positive correlation
with the genera *Bacillus*, *Sphingomonas*, *Massilia*, *Burkholderia-Caballeronia*-
*Paraburkholderia*, and *Cupriavidus*, while a negative correlation with the genus
*Pantoea*. Tax4Fun2 prediction indicated that T3 group exhibited significant enrichment
in pathways related to carbohydrate metabolism, amino acid metabolism, lipid

metabolism, metabolism of terpenoids and polyketides, and biosynthesis of other
secondary metabolites, suggesting that strain SCA4-21 could play an important role in
carbon and nitrogen metabolisms within the ecological environment.

Therefore, *S. luomodensis* SCA4-21 represents a potential bioagent for the control of
banana *Fusarium* wilt. This bioagent may achieve this by producing antifungal
compounds, enriching beneficial bacteria and fungi, and simultaneously reducing the
abundance of *Fusarium*
**Materials and methods**

**Extraction and purification of strain SCA4-21 extracts**

The extracts of strain SCA4-21 were extracted and purified following the method of
Li et al. (10). Namely, strain SCA4-21 was inoculated into 1000 mL of SLM (20 g of
soluble starch, 15 g of soybean powder, 5 g of yeast powder, 2 g of peptone, 4 g of
CaCO₃, 4 g of NaCl, 1 L of ddH₂O, pH 7.2–7.4) and cultured at 28°C for 7 days at
180 r / min. Then the fermentation broth was added 95% ethanol in a 1:1 ratio and
shocked at 28 °C and 180 r / min for 2 days. The mixtures were filtered by filter
paper, and the filtrate was concentrated by rotary vacuum evaporator (N-1300,
BoLang Co., Ltd., Shanghai, China). To separate and purify extracts, the concentrate
was fully adsorbed onto macroporous resin (D101) and eluted sequentially with 50%,
60%, 70%, 80% and 100% methanol gradient. These eluents were evaporated dry
using rotary vacuum evaporator, respectively. These separated and purified extracts
were dissolved in 10% of dimethyl sulfoxide (DMSO) with 20 mg/mL of a
concentration, filtered using a 0.22- μm sterile filter (Millipore, Bedford, MA, United
States) and preserved at -4 °C, respectively.

The antifungal activity of the extracts against Foc TR4 was determined using agar
dilution method (50). Briefly, 100 μL of extract solutions from different eluents were

added to 60 mL of PDA medium that had been sterilized but not yet solidified,
respectively. Every 60mL of PDA medium is evenly poured into three petri dishes to
make plates. Three plates with 10% of DMSO were used as controls. Each plate in the
center was inoculated a Foc TR4 mycelial plug with 0.5 cm diameter and incubated at
28 °C for 7 days. The clonal diameter of Foc TR4 was measured and fungal growth
inhibition (FGI) was calculated using the following formula: $FGI = [(C-T)/C]$
$\times 100\%$, where C and T represented the growth diameters of the control and the
treatment group, respectively. The most effective extracts were selected for further
investigations.

**Measuring the EC₅₀ of strain SCA4-21 extracts**

PDA plates with final concentration extracts of 0.78, 1.56, 3.13, 6.25, 12.50, 25.00,
50.00, 100.00, 200.00 µg/ml respectively were made up. The fungal disk of Foc TR4
(0.5cm diameter) was inoculated on PDA plates and cultivated in an incubator at 28°C
for 7 days. The plates with DMSO (10%, v/v) were used as a control. Three biological
replicates for each concentration were performed. The growth diameter of Foc TR4 was
measured, and the percentage of mycelial inhibition was calculated. A linear regression
equation was established using the least square method to determine the half maximal
effective concentration (EC₅₀) of extracts on the mycelial growth of Foc TR4 (16).

**2.3. Determining broad-spectrum antifungal activity of strain SCA4-21 extracts**

The broad-spectrum antifungal activity of strain SCA4-21 extracts against eight plant
pathogenic fungi were determined using agar well diffusion method (51). The eight
pathogenic fungi include *Curvularia fallax* (ATCC 34598), *Colletotrichum*
*gloeosporioides* (Penzig) Penzig et Saccardo (ATCC MYA-456), *Colletotrichum*
*gloeosporioides* (ACCC 36351), *Colletotrichum acutatum* (ATCC 56815),
*Colletotrichum fragariae* (ATCC 58718), *Colletotrichum gloeosporioides* (Penz) Saec

(ACCC 36351), *Fusarium oxysporum* f. sp. *cucumerinum* Owen. anamorph (ATCC
204378), *Fusarium graminearum* (DSM 21803). The pathogens were provided friendly
by the Institute of Environment and Plant Protection, China Academy of Tropical
Agricultural Sciences, Haikou, China. Fungal growth inhibition (FGI) was calculated
as above formula.

**2.4. Effect of strain SCA4-21 extracts on spore germination of Foc TR4**

Effect of strain SCA4-21 extracts on spore germination of Foc TR4 was observed
using our previous method (52) with minor modifications. 5000 mL of sterile water
was added to the well-growing Foc TR4 plate, and the mycelia and spores were
scraped using the sterilized coating rod. The spore suspension obtained by filtering off
mycelia was diluted to 1×10^6 CFU / ml and mixed with equal volume of different
concentration extracts ($1 \times EC_{50}$, $2 \times EC_{50}$, $4 \times EC_{50}$, $8 \times EC_{50}$). 10% DMSO was
used as control instead of extracts. After incubate at 28°C for 24h, The spore
germination of Foc TR4 was observed by an inverted microscope (Cellcutplus, MMI,
Germany), 100 conidia of each field were counted. The spore germination inhibition
(SGI) was calculated following the formula: $SGI = (A - B)/A \times 100\%$, where A and B
signify the spore germination rate of the control group and the treatment group,
respectively.

**Effect of strain SCA4-21 extracts on mycelial and spore morphology as well as the** 540 **ultrastructure of Foc TR4**

Effect of strain SCA4-21 extracts on mycelial morphology of Foc TR4 was observed
following the method of Cao et al. (53), with some modification. PDA plates containing
$4 \times EC_{50}$ extracts of strain SCA4-21 were utilized as treatments, while PDA plates
containing 10% DMSO served as controls. Disks (0.5cm-diameter) of Foc TR4 were
inoculated at the centers of these plates. After 3 days of culture, fungal mycelia blocks

(0.5-cm diameter) were cut from the edge of pathogen clone, fixed with 2.5% (v/v)
glutaraldehyde (C₃H₈O₂) at -4 °C overnight, and rinsed twice using phosphate-buffered
saline (0.1 mol/L, pH 7.4). Then, those fungal mycelia blocks underwent gradient
dehydration in 30%, 50%, 70%, 80%, 90%, 95% and 100% ethanol for 2 min each,
followed by a rinse with tert-butanol for 20 min. These fungal mycelium blocks were
subsequently soaked in fresh tert-butanol and frozen at -80 °C. They were then
subjected to freeze-drying using a freeze dryer. (FDU-2110, EYELA, Tokyo, Japan).
The sample sheet was fixed on a small steel column and coated with a film of gold-
palladium alloy under vacuum. Then the morphology of the hypha was observed by
scanning electron microscopy (SEM, Zeiss ΣIGMA, Germany). Fungal hyphae were
also collected from the above 3-day-old treatment and control plates, respectively and
treated with according to our previous method (53). The ultrastructure of fungal hyphae
was detected by transmission electron microscopy (JEM-1400Flash, Japan).
In addition, Foc TR4 spores suspension with 1×10⁶ CFU concentration was prepared
according to above method and mixed with 4×EC₅₀ extracts solution of strain SCA4-
21 (v:v=1:1). 20 μL of mixture was dropped onto a cover plate which was placed on a
glass slide and cultivated for at 28 °C 24 h. Control was set up by replacing extracts
solution with 10% DMSO. Spores were treated as described above method and
observed by SEM (Zeiss ΣIGMA, Germany).

**Identification of the volatile organic compounds of strain SCA4-21**

To identify the volatile organic compounds of strain SCA4-21, one hundred microliters
of strain SCA4-21 suspension (10⁶ CFU mL⁻¹) was inoculated into a sterilized 50-mL
flask containing 15 mL of SLM. Equal volume of ddH₂O replacing strain SCA4-21
suspension was used as control. Both treatment and control were set up with three
replicates each. After being cultured for 7d at 28°C with agitation at 180 rpm, the

volatile organic compounds of strain SCA4-21 were analyzed using GC-MS
(Clarus690+SQ8T, PerkinElmer, USA) following the method described in our previous
study (53) and identified by comparing the mass spectra of treatments with those of
controls.

**Evaluation of the biocontrol efficiency of strain SCA4-21 and its promotion effect**
**on banana seedlings**

The pot experiment of banana seedlings was carried out to assess biocontrol potential
of strain SCA4-21 against banana fusarium wilt. GFP-Foc TR4 strain expressing
green fluorescent protein (GFP) was provided kindly by the Institute of Tropical
Bioscience and Biotechnology, China Academy of Tropical Agricultural Sciences,
Haikou, China. The spore suspension of GFP-Foc TR4 was produced by inoculating
the pathogen in potato dextrose broth medium (PDB) and shaking (180 rpm) at 28°C
for 7 days, and adjusted to final concentration of 1×10^6 CFU/mL. The roots of healthy
banana seedlings with 3-4 leaves were wound with scissors and immersed in the spore
suspension of GFP-Foc TR4 for 30 min, then the seedlings were planted in plastic pots
filled with 900 g of soils to use as positive control or treatment. Healthy banana
seedlings untreated with GFP-Foc TR4 were planted as negative control. Namely,
three experimental groups were designed: T1 (sterilized soybean liquid medium:
SLM), T2 (Foc TR4+ SLM) and T3 (Foc TR4+SCA4-21). Each treatment group is 30
pots. The fermentation broth of strain SCA4-21 derived from inoculating the isolates
into SLM and shaking (180 rpm) at 28°C for 7 days. T1 group seedlings were watered
with 100 mL of SLM diluted 10 times once a week. T2 and T3 group seedlings were
irrigated with spore suspension of GFP-Foc TR4 at concentration of 1×10^6 CFU/mL
and 100 ml of SLM or fermentation broth diluted 10 times once per week.

After banana seedlings planted in a greenhouse at 28°C, with 70% of humidity and 12

596 h dark/12 h natural light for 60 days, the corms and roots of the banana seedlings were
597 cut to observe the infection degree of Foc TR4 using laser confocal microscope (Axio
Scope A1, Carl ZEISS, Germany). The physiological and biochemical indexes of
banana plants under experimental group were determined, including chlorophyll
content, leaf area, plant height, stem diameter, and biomass (dry weight, fresh weight)
at 120 th day. The disease index and biocontrol efficiency were calculated according
to the method of Chen et al. (15).

**Effect of strain SCA4-21 on the microbial community structure of banana** 604 **rhizosphere soil**

To decipher the effect of strain SCA4-21 on the microbial community structure of
banana rhizosphere soil, a total of 3 rhizosphere soil samples were collected for each
experimental group and stored in a -80°C. The collected samples were sent to Beijing
Baimaike Biotechnology Co., Ltd. for microbial diversity **sequencing using paired-end**
**method on Illumina NovaSeq sequencing platform**. Bioinformatics analysis was
performed using BMKCloud (www.biocloud.net). Raw reads were filtered using the
trimmomatic (v 0.33) software (54) and adapter sequences were removed to acquire
clean reads using cutadapt (v1.9.1) (55). Clean reads were denoised, spliced and
removed the chimeric sequences using the dada2 method in QIIME2 (v 2020.6)
software to obtain non-chimeric reads (56-57). Operational taxonomic units (OTUs)
were identified and Venn diagrams were made using QIIME2 (v 2020.6) software
(57). The bacterial and fungal alpha and beta diversity indices were analyzed using
QIIME2 (v 2020.6) (57). Specifically, Shannon, Simpson, Chao1 and ACE indices were
measured to estimate microbial alpha diversity. The Wilcoxon test was selected for
between-group variance analysis of alpha diversity indices at OUT level. Principal
components analysis (PCA) and nonmetric multidimensional scaling (NMDS) were

visualized to assess microbial beta diversity. The Binary-Jaccard distance metric was
chosen for analyzing between-group variance in the NMDS analysis. Taxonomic
annotation of OTUs was performed using SILVA database (58) ([http://www.arb-](http://www.arb-silva.de/)
[silva.de/](http://www.arb-silva.de/)). A community structure map at the level of microbial genera was created
using R language tools in QIIME2 (v 2020.6) software (57). Intergroup differences in
abundance of microbial genera levels were analyzed using analysis of variance
(ANOVA) method. Functional profiles of bacteria community were predicted using
Tax4Fun2 (57). Bacterial network analysis was performed using Molecular Ecological
Network Analyses Pipeline to elucidate the effects of different treatments on bacterial
ecological network (60) (<http://ieg4.rccc.ou.edu/mena>).

**Statistical Analysis**

Statistical analysis was performed with the SPSS Version 22.0 software (SPSS Inc.,
Chicago, IL, United States). Significant difference between means was determined by
the Duncan's multiple range test at $p < 0.05$. The difference among treatments was
determined using one-way analysis of variance (ANOVA). All data from three
biological replicates of each experiment were obtained as means \pm the standard error
(SE).

**Declaration of Competing Interest**

The authors declare that they have no competing interests.

**Data Availability**

The data that has been used is confidential.

**Acknowledgments**

This study was supported by the project of National Key Laboratory for Tropical Crop
Breeding (NKLTCB202306), the National Natural Science Foundation of China
(U22A20487 and 322MS126), The Natural Science Foundation of Hainan

(322QN417 and 322MS126), Chinese Academy of Tropical Agricultural Sciences for
Science and Technology Innovation Team of National Tropical Agricultural Science
Center (CATASCXTD202309 and CATASCXTD202312), Central Public-interest
Scientific Institution Basal Research Fund (1630052023011) and the China Agriculture
Research System (CARS-31).

**References**

- 1. Marta H, CahyanaY, Djali M, Prama fisi G. 2022. The Properties, modification, and
application of banana starch. *Polymers (Basel)* 14(15):3092. doi:
10.3390/polym14153092.
- 2. Perez-Vicente L, Dita M, Martinez DLPE. 2014. Technical manual prevention and
diagnostic of *Fusarium* wilt (Panama disease) of banana caused by *Fusarium*
*oxysporum* f. sp. *cubense* tropical race 4 (TR4). Food and Agriculture Organization
of the United Nations. Technical manual pp:1-75.
- 3. Dita M, Barquero M, Heck D, Mizubuti ESG, Staver CP. 2018. *Fusarium* wilt of
banana: current knowledge on epidemiology and research needs toward sustainable
disease management. *Front Plant Sci* 9:1468.
<https://doi.org/10.3389/fpls.2018.01468>.
- 4. Van Bergeijk DA, Terlouw BR, Medemam M H, van Wezel GP. 2020. Ecology and
genomics of Actinobacteria: new concepts for natural product discovery. *Nat Rev*
*Microbiol* 18(10):546-558. <https://doi.org/10.1038/s41579-020-0379-y>.
- 5. Weller DM. 1988. Biological control of soil borne plant pathogens in the
rhizosphere with bacteria. *Ann Rev Phytopathol* 26:379-407. [https://doi.org/](https://doi.org/10.1146/annurev.py.26.090188.002115)
[10.1146/annurev.py.26.090188.002115](https://doi.org/10.1146/annurev.py.26.090188.002115).
- 6. El-Tarabily KA, Soliman MH, Nassar AH, Al-Hassani HA, Hardy GESJ. 2000.
Biological control of *Sclerotinia minor* using a chitinolytic bacterium and
actinomycetes. *Plant Pathol* 49:573-583. [https://doi.org/10.1046/j.1365-](https://doi.org/10.1046/j.1365-3059.2000.00494.x)
[3059.2000.00494.x](https://doi.org/10.1046/j.1365-3059.2000.00494.x).
- 7. Tokala RK, Strap JL, Jung CM, Crawford DL, Salove MH, Deobald LA, Bailey JF,

- Morra MJ. 2002. Novel plant-microbe rhizosphere interaction involving
*Streptomyces lydicus* WYEC108 and the pea plant (*Pisum sativum*). *Appl Environ*
*Microbiol* 68 (5):2161-71. <https://doi.org/10.1128/AEM.68.5.2161-2171.2002>.
- 8. Kieser T, Bibb MJ, Buttner MJ, Chater KF, Hopwood DA. 2000. Practical
streptomyces genetics. John Innes Foundation pp:5-6.
<http://www.amazon.co.uk/exec/obidos/ASIN/0708406238/citeulike-21>.
- 9. Law JW, Ser HL, Khan TM, Chuah LH, Pusparajah P, Chan KG, Goh BH, Lee L H.
2017. The potential of *Streptomyces* as biocontrol agents against the rice blast
fungus, *Magnaporthe oryzae* (*Pyricularia oryzae*). *Front Microbiol* 8:3.
<https://doi.org/10.3389/fmicb.2017.00003>.
- 10. Li X, Jing T, Zhou D, Zhang M, Qi D, Zang X, Zhao Y, Li K, Tang W, Chen Y, Qi
C, Wang W, Xie J. 2021. Biocontrol efficacy and possible mechanism of
*Streptomyces* sp. H4 against postharvest anthracnose caused by *Colletotrichum*
*fragariae* on strawberry fruit. *Postharvest Biol Tec* 175:111401.
<https://doi.org/10.1016/j.postharvbio.2020.111401>.
- 11. Zhou D, Jing T, Chen Y, Yun T, Qi D, Zang X, Zhang M, Wei Y, Li K, Zhao Y, Wang,
690 W. 2022. Biocontrol potential of a newly isolated *Streptomyces* sp. HSL-9B from
691 mangrove forest on postharvest anthracnose of mango fruit caused by
692 *Colletotrichum gloeosporioides*. *Food Control* 135:108836.
<https://doi.org/10.1016/j.foodcont.2022.108836>.
- 12. Bérdy J. 2005. Bioactive microbial metabolites. *J Antibiotics* 58: 1-26.
<https://doi.org/10.1038/ja.2005.1>
- 13. Shepherdson EMF, Elliot MA. 2022. Cryptic specialized metabolites
drive *Streptomyces* exploration and provide a competitive advantage during growth
with other microbes. *Proc. Natl. Acad. Sci. USA*. 119(40):e2211052119.
<https://doi.org/10.1073/pnas.2211052119>.
- 14. Ningthoujam S, Sanasam S, Tamreihao K, Nimaich S. 2009. Antagonistic activities
of local actinomycete isolates against rice fungal pathogens. *Afr J Microbiol Res*
3:737-742.
- 15. Chen Y, Zhou D, Qi D, Gao Z, Xie J, Luo Y. 2018. Growth promotion and disease
suppression ability of a *Streptomyces* sp. CB-75 from banana rhizosphere soil *Front*
*Microbiol* 8:2704. <https://doi.org/10.3389/fmicb.2017.02704>.

- 16. Li X, Li K, Zhou D, Zhang M, Qi D, Jing T, Zang X, Qi C, Wang W, Xie J. 2021.
Biological control of banana wilt disease caused by *Fusarium Oxyspoum* f. sp.
*Cubense* using *Streptomyces* sp. H4. *Biol Control* 155:104524.
<https://doi.org/10.1016/j.biocontrol.2020.104524>.
- 17. Qi D, Zou L, Zhou D, Zhang M, Wei Y, Li K, Zhao Y, Zhang L, Xie J. 2022.
Biocontrol potential and antifungal mechanism of a novel *Streptomyces*
*sichuanensis* against *Fusarium oxysporum* f. sp. *cubense* tropical race 4 in vitro and
in vivo. *Appl Microbiol Biotechnol* 106(4):1633-1649.
<https://doi.org/10.1007/s00253-022-11788-3>.
- 18. Qi D, Liu Q, Zou L, Zhang M, Li K, Zhao Y, Chen Y, Feng J, Zhou D, Wei Y,
Wang W, Zhang L, Xie J. 2024. Taxonomic identification and antagonistic activity
of *Streptomyces luomodiensis* sp. nov. against phytopathogenic fungi. *Front*
*Microbiol* 15:1402653. <https://doi.org/10.3389/fmicb.2024.1402653>.
- 19. Behera HT, Mojumdar A, Behera SS, Das S, Ray L. 2022. Biocontrol of wilt disease
of rice seedlings incited by *Fusarium oxysporum* through soil application of
*Streptomyces chilikensis* RC1830. *Lett Appl Microbiol* 75(5):1366-1382. doi:
10.1111/lam.13807.
- 20. Chen Y, Che, P, Tsay T. 2016. The biocontrol efficacy and antibiotic activity of
*Streptomyces plicatus* on the oomycete *Phytophthora capsici*. *Biol Control* 98:
34-42. <https://doi.org/10.1016/j.biocontrol.2016.02.011>.
- 21. Wei Y, Zhao Y, Zhou D, Qi D, Li K, Tang W, Chen Y, Jing T, Zang X, Xie J, Wang
727 W. 2020. A Newly Isolated *Streptomyces* sp. YYS-7 With a broad-spectrum
antifungal activity improves the banana plant resistance to *Fusarium oxysporum* f.
sp. *cubense* tropical race 4. *Front Microbiol* 11:1712.
<https://doi.org/10.3389/fmicb.2020.01712>.
- 22. Khan MA, Ahmad R, Srivastava AN. 2016. Effect of methyl butyrate aroma on the
survival and viability of human breast cancer cells in vitro. *J Egypt Natl Canc Inst*
28(2):81-88. <https://doi.org/10.1016/j.jnci.2016.02.005>.
- 23. Ayaz M, Ali Q, Farzand A, Khan AR, Ling H, Gao X. 2021. Nematicidal volatiles
from *Bacillus atrophaeus* GBSC56 promote growth and stimulate induced systemic
resistance in tomato against *Meloidogyne incognita*. *Int J Mol Sci* 22(9):5049.
<https://doi.org/10.3390/ijms22095049>.
- 24. Morita T, Tanaka I, Ryuda N, Ikari M, Ueno D, Someya T. 2019. Antifungal

- spectrum characterization and identification of strong volatile organic compounds
produced by *Bacillus pumilus* TM-R. *Heliyon* 5(6):e01817.
<https://doi.org/10.1016/j.heliyon.2019.e01817>.
- 25. Zhang D, Qiang R, Zhao J, Zhang J, Cheng J, Zhao D, Fan Y, Yang Z, Zhu J. 2022.
Mechanism of a volatile organic compound (6-Methyl-2-heptanone) emitted
from *Bacillus subtilis* ZD01 against *Alternaria solani* in potato. *Front Microbiol*
12:808337. <https://doi.org/10.3389/fmicb.2021.808337>.
- 26. Dotson BR, Verschut V, Flardh K, Becher PG, Rasmusson AG. 2020. The
*Streptomyces* volatile 3-octanone alters auxin/cytokinin and growth in *Arabidopsis*

[revised manuscript text omitted]

2105-13-113.

**Figure Legends**

- 1. Isolated and activity assay of *S. luomodensis* SCA4-21 extracts. (A) Effect of
extracts separated by different gradient methanol solvents on the mycelial growth
of Foc TR4. (B) Antifungal evaluation of extracts at different concentrations on the
mycelial growth of Foc TR4. (C) Antifungal evaluation of extracts on the mycelial
growth of other eight phytopathogenic fungi.
- 2. Effect of extracts on spore germination, spore and hyphal morphology as well as
ultrastructure of Foc TR4 cells. (A) Inhibitory effects of extracts at different
concentrations on the spore germination of Foc TR4. (B) Quantitative analysis of
germination rate. Different letters represented a significant difference at the $P < 0.05$
level by the Duncan's multiple range test. (C) Morphological characteristics of Foc
TR4 hypha treated with $4 \times EC_{50}$ of extracts. (D) Morphological characteristics of
Foc TR4 spores treated with $4 \times EC_{50}$ of extracts. (E) Ultrastructural characteristics
of Foc TR4 cell treated with $4 \times EC_{50}$ of extracts. The control groups were treated
with 10% DMSO.
- 3. Chemical structures of the identified compounds of *S. luomodensis* SCA4-21 by
GC-MS.
- 4. Biocontrol efficacy of *S. luomodensis* SCA4-21 against banana *Fusarium* wilt and
promotive effect on growth of banana seedlings. (A) In vivo effects of *S.*
*luomodensis* SCA4-21 on banana *Fusarium* wilt. (B) Quantitative analysis of
disease index. Quantitative analysis of growth-related indices, including plant
height (C), stem diameter (D), chlorophyll (E), leaf area (F), vane thickness (G),
and biomass (H). Different lowercase letters indicated a significant difference

among different treatments according to the Duncan's multiple range test ($p < 0.05$).

5. Effect of *S. luomodiensis* SCA4-21 on diversity of rhizosphere soil bacterial
communities. (A) Identification of OTUs in different soil samples. (B) Venn
diagram of different treatment. (C, D) Diversity indices Shannon and Simpson. (E,
F) Richness indices Chao1 and ACE. (G) Principal component analysis of different
treatment. (H) Nonmetric multidimensional scaling analysis of different treatment.

6. Effect of *S. luomodiensis* SCA4-21 on composition of rhizosphere soil bacterial
communities. (A) Relative abundances of top 20 bacterial genus under each
treatment. (B) Analysis of intergroup differences in the abundance of the top 20
bacterial genera using the analysis of variance (ANOVA) method.

7. Effect of *S. luomodiensis* SCA4-21 on function and correlation of rhizosphere soil
bacteria. (A) Functional profile prediction of bacteria community using Tax4Fun2.
(B) Network correlation analysis of top 20 bacterial genus.

8. Effect of *S. luomodiensis* SCA4-21 on diversity of rhizosphere soil fungal
communities. (A) Identification of OTUs in different soil samples. (B) Venn
diagram of different treatment. (C, D) Diversity indices Shannon and Simpson. (E,
F) Richness indices Chao1 and ACE. (G) Principal component analysis of different
treatment. (H) Nonmetric multidimensional scaling analysis of different treatment.

9. Effect of *S. luomodiensis* SCA4-21 on composition of rhizosphere soil fungal
communities. (A) Relative abundances of top 20 bacterial genus under each
treatment. (B) Analysis of intergroup differences in the abundance of the top 20
bacterial genera using the analysis of variance (ANOVA) method.

Table Legends

Table 1 Statistics of bacterial 16S rRNA gene sequencing data processing results for soil samples with different treatments

Table 2 Bacterial species identified at different levels in soil samples under different treatment

Table 3 Statistics of fungal internal transcribed spacer sequencing data processing results for soil samples with different treatments

Table 4 Fungal species identified at different levels in soil samples under different treatments

982

Table 1

983

Statistics of bacterial 16S rRNA gene sequencing data processing results for soil

984

samples with different treatments

Sample ID	Raw Reads	Clean Reads	Denosed Reads	Merged Reads	Non-chimeric Reads
SLM	79966	79771	74069	59601	50140
SLM	80121	79953	74192	58275	47570
SLM	79926	79767	74365	60998	51339
Foc TR4+SLM	80039	79878	74308	66628	59763
Foc TR4+SLM	80123	79937	73295	60729	51707
Foc TR4+SLM	79909	79735	72242	59815	51019
Foc TR4+SCA4-21	80362	80187	74852	68004	58432
Foc TR4+SCA4-21	79863	79684	74327	70695	62599
Foc TR4+SCA4-21	79878	79682	73340	62188	49566
Total	720187	718594	664990	566933	482135

Table 2

Bacterial species identified at different levels in soil samples under different treatment

Sample	Kindom	Phylum	Class	Order	Family	Genus	Species
SLM	2	33	73	191	344	533	576
SLM	1	26	67	157	282	430	457
SLM	1	27	63	167	290	428	459
Foc TR4+SLM	1	28	61	155	265	417	456
Foc TR4+SLM	2	33	70	176	312	526	581
Foc TR4+SLM	1	27	58	145	250	383	420
Foc TR4+SCA4-21	1	27	58	155	252	374	407
Foc TR4+SCA4-21	1	28	63	161	269	408	431
Foc TR4+SCA4-21	1	27	66	167	270	394	425
Total	2	40	99	265	515	971	1142

Table 3

Statistics of fungal internal transcribed spacer sequencing data processing results

for soil samples with different treatments

Sample ID	Raw Reads	Clean Reads	Denoisied Reads	Merged Reads	Non-chimeric Reads
SLM	79906	79566	78867	77548	75255
SLM	79890	79505	78644	77148	77023
SLM	80044	79637	78167	76089	74962
Foc TR4+SLM	80085	79814	79456	78027	77692
Foc TR4+SLM	80424	80067	79277	77274	77216
Foc TR4+SLM	80110	79831	79108	76998	76793
Foc TR4+SCA4-21	80324	79973	79605	77926	77814
Foc TR4+SCA4-21	79984	79581	79301	78148	77432
Foc TR4+SCA4-21	80031	79698	79056	76523	75989
Total	720798	717672	711481	695681	690176

Table 4

Fungal species identified at different levels in soil samples under different treatments

Sample	Kindom	Phylum	Class	Order	Family	Genus	Species
SLM	1	9	24	53	85	106	111
SLM	1	8	26	59	93	133	144
SLM	1	10	32	72	132	200	227
Foc TR4+SLM	1	8	24	59	92	131	142
Foc TR4+SLM	1	10	29	69	111	139	142
Foc TR4+SLM	1	10	27	61	97	129	136
Foc TR4+SCA4-21	1	11	30	70	110	166	196
Foc TR4+SCA4-21	1	9	29	54	98	140	160
Foc TR4+SCA4-21	1	10	26	65	120	173	189
Total	1	11	42	105	217	400	537

1000

1001

$$EC_{50}=37.88 \pm 1.58 \text{ (}\mu\text{g/mL)}$$

Butanoic acid, methyl ester (Methyl butyrate)

Ethene, fluoro-

Methyl isovalerate

Pentanoic acid, 3-methyl-, methyl ester

Pentanoic acid, 4-methyl-, methyl ester

Hexanoic acid, methyl ester (methyl hexanoate)

2-Heptanone, 6-methyl-

2-Heptanone, 5-methyl-

(E)-2-Hexenoic acid, methyl ester (Methyl (E)-2-hexenoate)

3-Octanone

Hexanoic acid, 5-methyl-, methyl ester

Hexanoic acid, 2-ethyl-, methyl ester (Methyl 2-ethylhexanoate)

Benzoic acid, methyl ester (methyl benzoate)

2-Octenoic acid, methyl ester, (E)-

2-Methylisoborneol

3-(1,3,5-Cycloheptatrien-7-yl)-2,4-pentanedione

Comments by the Editor

1. Methods should be described in more detail, especially the VOC experiment, sequencing and bioinformatics analyses.

Response to comments: Thanks for the editor's suggestion. The detailed procedures for the VOC experiment, microbial diversity sequencing, and bioinformatics analyses have been thoroughly elaborated in the revised manuscript (please see Pages 584–608, and 650–693 for full details).

2. ANOVA is not suited for compositional data so you may want to explore other alternative statistics for your metabarcoding data (ANCOM for example).

Response to comments: We thank the editor for this valuable suggestion. We have now re-analyzed our metabarcoding data using ANCOM-BC, which has strengthened our statistical analysis. The revised results, which confirm our main findings, are now detailed in the manuscript (lines 255-257, 319-321, 679-681, as well as Figs. 6B and 9B).

3. A proper control in the in vivo experiment is lacking ("healthy plants" should have been immersed as well for 30 min).

Response to comments: We sincerely apologize for the lack of clarity in our original description, which led to a misunderstanding. To improve clarity, we have now revised the manuscript to explicitly detail the handling procedures for all groups, including the physical treatment of the negative controls (please see lines 621–636 for full details) .

4. The rationale behind some experiments should be better introduced. For example, why testing the extracts against pathogens that are not soil-borne? Why not testing the extracts in the in vivo experiment?

Response to comments: We thank the editor for this insightful comment regarding the rationale of our experimental design. We agree that providing a clearer explanation is necessary and have revised the manuscript accordingly. The modifications can be found in the Introduction (Lines 103-115) and the Materials and Methods section (Lines 518-521 and 611-613).

Minor comments:

1. L53: Link the first sentence and the second. The switch to the description of the

pathogen is rather abrupt. Maybe some data regarding the incidence of the pathogen, and its worldwide relevance?

Response to comments: We thank the editor for this helpful suggestion. We have now incorporated the provided sentence with key epidemiological and economic impact data for *Fusarium oxysporum* f. sp. *cubense* (Foc TR4) in lines 53-57 to seamlessly link the general threat of fungal diseases to the specific pathogen and emphasize its global relevance.

2. L54: "Foc TR4 is susceptible to almost all banana cultivars". The other way around?

Response to comments: Thanks for the editor's correction. We have revised the sentence accordingly. Please see lines 57-58.

3. L61: alternative to what?

Response to the comment: We thank the editor for this helpful comment. We have clarified the sentence on line 65 to specify that biocontrol is an alternative to chemical control methods.

4. L65-75: please focus on soil-borne pathogens to be consistent with your previous sentences about rhizosphere *Streptomyces* for the control of soil-borne pathogens. Not all the cited studies refer to rhizospheric strains.

Response to comments: We thank the reviewer for this valuable comment. We have revised the paragraph (lines 66-74) to focus exclusively on soil-borne pathogens and ensure all cited studies are consistent with the scope of rhizosphere-associated strains. The examples of phytopathogens and corresponding references (now on lines 793-816) have been updated accordingly.

5. L71: "found", not "founded"

Response to the comment: We thank the editor for this helpful feedback. We have performed a thorough grammatical check and revised the manuscript accordingly. The specific term "founded" mentioned by the editor is no longer present in the revised text .

6. L77: *in vitro* in italics

Response to the comment: We thank the editor for pointing this out. "*In vitro*" has been italicized in the revised manuscript as recommended (now on line 93).

7. Rather than a description of the results in the Introduction, I would rather prefer to see some background regarding the potential of rhizosphere strains as biocontrol agents of FOC TR4 in banana and a more in depth description of studies cited as 15, 16 and 17 (antagonistic activity of *Streptomyces* strains against FOC TR4). This would help emphasise the relevance of the present study. Please provide a clear objective for this study instead of enlisting the experiments you performed in the Introduction.

Response to the comments: Thank you for your constructive comments. We have revised the Introduction accordingly. We have now expanded the background to include a more in-depth discussion on the antagonistic mechanisms of our previously reported rhizosphere *Streptomyces* strains (now references 16–18). This provides a stronger foundation for the present study. Additionally, as requested, we have replaced the experimental list with a clear statement of objectives (please see lines 79-105).

8. L500: colonial instead of clonal?

Response to the comment: Thank you for pointing out this critical typographical error. We have corrected "clonal" to "colonial" on line 502.

9. L504: did you consider using a positive control for the antifungal activity, for example a commercial fungicide used in the treatment of FOC TR4?

Response to the comment: We thank the reviewer for this excellent suggestion. We have now included tebuconazole, a fungicide commonly used against *Fusarium* spp., as a positive control. Tebuconazole was dissolved in 10% DMSO and incorporated into PDA medium at a final concentration of 33.3 µg/mL. The positive control plates were prepared in triplicate and all assays were performed alongside the experimental treatments. The results have been added to Figure 1A and the methodology has been described in the Materials and Methods section (lines 499-500).

10. L514: The statement that the extracts were tested against 8 pathogenic fungi is misleading, since some of these fungi are the same species (albeit different strains). What was the purpose of testing the extracts against these fungi? Some of them are not soil-borne, so this should be justified. Are they all affecting banana?

Response to the comments: We thank the editor for catching this oversight. The text has

been corrected to clarify that the extracts were tested against eight pathogenic isolates representing six species. This was done to assess the broad-spectrum activity of the extract against a panel of economically important pathogens (please see lines 518-521, 531-534).

11. Replicates for section 2.4?

Response to the comment: We thank the editor for catching this oversight. The experiments were performed with three biological replicates, and this information has been explicitly stated in the revised manuscript (please see Lines 548-549).

12. L533: "incubating", not "incubate".

Response to the comment: Thank you for pointing out this error. The verb form has been corrected to "incubating" in the revised manuscript (Line 549).

13. L542: how many PDA plates?

Response to the comment: We thank the editor for this valuable comment. We agree that specifying the number of biological replicates is crucial for demonstrating the reproducibility of the results. We have revised the manuscript to explicitly state that three replicate plates were used for both the treatment and the control groups (please see lines 559-560).

14. L567: were inoculated

Response to comment: We thank the editor for raising this point. To improve clarity and avoid any potential ambiguity, we have revised the sentence as follows: "A volume of 100 μL of strain SCA4-21 suspension (10^6 CFU mL^{-1}) was inoculated..." (please see lines 587-588).

15. Please describe the methods for capturing the VOCs in more details: type of SPME fibers? Parameters for the GC-MS analysis? Compound database? I understand it was described in another paper but details should be added here as well.

Response to the comments: We thank the editor for this suggestion. We have now added the requested details to the Methods section (please see lines 592-608).

16. Please ensure the titles for subsections of the Methods are homogenized (some sections are numbered, others are not).

Response to the comments: We thank the editor for pointing out the inconsistency in the

subsection headings. We have now homogenized the format by removing the numbering from the method subheadings throughout the Materials and Methods section. The revisions can be seen on lines 517 and 541 of the revised manuscript.

17. The choice of inoculating the strain and not applying compounds (either extracts or VOCs) for biocontrol should be justified. The whole manuscript is aimed at determining the metabolites produced by this *Streptomyces* strain so the focus of the *in vivo* biocontrol experiment seems rather odd.

Response to the comments: We thank the editor for this insightful comment. The justification for applying the live strain, rather than purified compounds, in the *in vivo* experiment has now been added to the manuscript (Lines 103-115). Briefly, this approach was chosen to evaluate the comprehensive and sustainable biocontrol capacity of the strain under realistic conditions, which depends on the synergistic actions of the live bacterium in the rhizosphere.

18. L586: control for the *in vivo* experiment should have been immersed in dH₂O for 30 min to consider the effect of the bathing stress.

Response to the comments: Thank you for this excellent suggestion. We agree that immersing the control plants in dH₂O is important to account for the potential stress of the bathing process itself. We have now revised the manuscript accordingly on lines 621-636. The control plants were treated identically to the inoculated group, including the 30-minute immersion in sterile dH₂O.

19. L608: how was DNA extraction performed? How were the sequencing library prepared? Which primers were used? What was the sequencing depth?

Response to the comments: We thank the editor for raising these important methodological points. All the requested details are indeed provided in the 'Materials and Methods' section of our manuscript (specifically, lines 650-665).

20 . L614: Why OTUs and not ASVs?

Response to the comment: We thank the editor for this important observation. We apologize for the error in our initial manuscript. Upon verification with our sequencing provider, we confirm that the bioinformatic analysis was indeed performed using

the Amplicon Sequence Variants (ASVs) method, not OTUs. The term "OTUs" was used incorrectly throughout the text due to an oversight on our part.

We will correct this error in lines 670 and anywhere else it occurs to consistently use "ASVs," ensuring the terminology accurately reflects the methods used. We appreciate editor's vigilance in helping us improve the accuracy of our manuscript.

21. L619: OTUs, not OUT

Response to the comment: We sincerely thank the editor for catching this error. we have now uniformly updated our methodology throughout the manuscript to correctly reflect that our analysis was based on ASVs, not OTUs. This correction has been made on line 676 of the revised manuscript.

22. L627: ANOVA is not suited for compositional data.

Response to the comment: We thank the editor for this correct and important comment. We agree that standard ANOVA is not appropriate for the analysis of compositional data like microbiome relative abundances. In our revised manuscript, we have replaced all relevant statistical analyses with the ANCOM-BC method, which is specifically designed for identifying differentially abundant taxa in compositional datasets. The results and figures (e.g., Fig.6B and Fig.9B) have been updated accordingly. Please see Lines 679-681 in the revised manuscript for a description of the method.

23. What was the purpose of the bacterial network analysis? Networks based on correlations of taxa relative abundance may not reflect real ecological interactions. Please provide parameters for the network construction.

Response to the comments: We thank the reviewer for these critical questions regarding the co-occurrence network analysis. Please find our point-by-point response below:

(1)Purpose of the analysis: The primary purpose was to generate ecological hypotheses about the potential relationships (e.g., cooperation or competition) between the inoculated *Streptomyces* strain and the native bacterial community, moving beyond taxonomic composition to infer potential interactive dynamics that may contribute to the observed biocontrol effect (lines 275-281,380-391).

(2) Acknowledgment of limitation: We fully agree with the editor that correlations in relative abundance do not equate to direct, causal ecological interactions. We have now explicitly acknowledged this limitation in the Discussion section (Lines 380-391, framing our findings as hypothesized interactions that require future validation).

(3) Network parameters: The detailed parameters for network construction have been thoroughly provided in the Methods section (Lines 275-281).

24. Fig. 2C: could images with the same scale could be provided for better comparison?

Response to the comment: We thank the editor for the suggestion. As requested, we have replaced the images in Fig. 2C to ensure they are all presented at the same scale. The revised figure now allows for a better visual comparison.

25. L105: "effect" instead of "manner"?

Response to the comment: We thank the reviewer for this helpful suggestion. We have replaced "manner" with "effect" (please see Line 135).

26. Fig. 6A and 6B are redundant? I understand they present the same data. Presenting histograms at the genus level makes them hard to interpret, maybe at the class level? Differentially enriched genera could be emphasised in another way, through DeSeq for example. Same comment for the fungal community.

Response to the comments: We thank the editor for these valuable suggestions. We have revised the figures to enhance clarity: (1) To avoid redundancy and improve interpretation, Fig. 6A/9A now display the community composition at the class level, while Fig. 6B/9B remain as heatmaps at the genus level. (2) To emphasize differentially enriched genera, we have used ANCOM-BC—a method specifically robust for compositional microbiome data—with significant genera clearly marked by asterisks on the heatmaps (Figs. 6B and 9B).

27. L206: there was quite a lot of dispersion in my opinion. What makes you say that "Principal component analysis (PCA) revealed a tendency for different samples of the same treatment to cluster together"? Same comment for the fungal community.

Response to the comments: We sincerely thank the editor for this insightful comment. Upon re-examining the PCA plots, we agree that the original description did not accurately reflect the observed patterns. In response to this comment, we have thoroughly revised the descriptions of both the bacterial and fungal PCA results in the manuscript to provide a more precise and objective account (please see Lines 235-244 and 301-308).

28. Please do not enlist all relative abundance percentages as it makes the reading quite tedious. Emphasise your main results and differences between treatments.

Response to the comments: We sincerely thank the editor for this valuable suggestion. We completely agree that emphasizing the main findings and treatment differences greatly improves the clarity and impact of the manuscript. In direct response to this comment, we have thoroughly revised the results sections concerning both the bacterial and fungal communities across the entire manuscript (please see Lines 249-264 and 312-332).

29. L319: *in vitro* in italics

Response: We thank the editor for pointing this out. The term "in vitro" has now been italicized in the revised manuscript as requested (please see Line 337).

30. L339-342: why not focus on antifungal activities exhibited by these VOCs? You should acknowledge that the VOC profile of *Streptomyces* is likely to change when in contact with the pathogen or the soil native microbiota.

Response: We thank the editor for this valuable suggestion. As recommended, we have revised the manuscript to place a stronger emphasis on the antifungal activities of the identified VOCs. The updated text now explicitly highlights several key VOCs with known antifungal functions and cites relevant literature, while also acknowledging other biological activities in a secondary capacity. We agree that the VOC profile may change in different ecological contexts, and this important point will be considered in our future research (please see Lines 351-362).

31. L374-379: not surprising since correlation networks are based on your relative abundance results. Yet these are correlations and do not mean that there is a regulatory

effect of your beneficial strain on these taxa, please rephrase. These positive correlations may be due to other factors.

Response: We thank the editor for this insightful comment. We fully agree that correlation does not imply causation. Accordingly, we have rephrased the relevant paragraph (Lines 380-391) to remove the claim of a "regulatory effect" and to more cautiously discuss the observed correlations as potential microbial interactions that require future validation.

32. The predicted functions (Tax4fun) are not discussed. Why ?

Response: We thank the editor for this insightful comment. We fully agree that discussing the predicted functional implications is crucial. In response, we have now added a comprehensive discussion on the Tax4Fun predictions in the revised manuscript . The discussion interprets the significant upregulation of key metabolic pathways (e.g., biosynthesis of secondary metabolites, carbohydrate metabolism) and links these functional shifts to the observed biocontrol and plant growth promotion effects, with supporting citations included (please see Lines 409-421).

Reviewer #3 (Comments for the Author):

This manuscript presents a well-designed and comprehensive study investigating the antifungal efficacy and biocontrol mechanisms of *Streptomyces luomodiensis* SCA4-21 against *Fusarium oxysporum* f. sp. *cubense* TR4 (Foc TR4). The experimental framework is sound, and the integration of physiological, microbiological, metabolomic, and molecular data greatly strengthens the validity of the conclusions.

Strengths

1. The identification and functional characterization of a new *Streptomyces* species with potent antifungal activity is timely and highly relevant to managing banana *Fusarium* wilt.

Response: We thank the reviewer for this positive comment and for acknowledging the relevance and timeliness of our work.

2. The combined use of GC-MS, high-throughput sequencing, and plant growth

experiments provides a holistic view of the biocontrol mechanism.

Response: We appreciate the reviewer's recognition of our multi-faceted approach to elucidate the biocontrol mechanisms.

3. The beneficial effects on the rhizosphere microbiome and plant health are clearly demonstrated and supported by both qualitative and quantitative data.

Response: We are grateful to the reviewer for this supportive comment on the strength of our data.

Improvements:

1. Statistical Rigor and Transparency: Clarify assumptions for ANOVA tests (e.g., normality, homogeneity of variance).

Response: We thank the reviewer for raising this crucial point regarding statistical rigor. In response to this comment and to ensure the most robust identification of differentially abundant taxa, we have re-analyzed our high-throughput sequencing data using the ANCOM-BC method throughout the manuscript, instead of relying on ANOVA-based approaches (please see Lines 255-264, 319-332, 679-681, Fig.6B and Fig.9B).

The ANCOM-BC methodology was specifically developed for microbiome compositionality data and does not rely on the same assumptions of normality and homoscedasticity (homogeneity of variance) that are required for ANOVA. By adopting this more advanced and appropriate method, we have effectively addressed the underlying concern about statistical assumptions, thereby strengthening the validity and reliability of our conclusions regarding microbial community shifts.

2. Report exact p-values or significance indicators (ns, *, **) in all figure panels.

Response: We thank the reviewer for the suggestion. We have updated Figures 6 and 9 to include significance indicators. Specifically, the panels now show results at the class level (A) and present genus-level differences as heatmaps (B), with ANCOMBC results annotated using *, **, ***, and ns based on standard significance levels. These changes improve clarity and statistical reporting (Fig.6B and Fig.9B).

4. Clarify replicates: Define biological vs. technical replicates explicitly.

Response: We thank the reviewer for raising this important point. To clarify, the distinction between biological and technical replicates was indeed provided in the Methods section. As requested, we have now explicitly defined these terms in the revised manuscript to ensure maximum clarity (please see Lines 500-501, 526-527, 548-549, 589-590, and 653-654).

5. Indicate post-hoc test results in figure legends (e.g., Duncan's test letters).

Response: We thank the reviewer for this suggestion. We have now explicitly indicated the results of the post-hoc tests in the legends of Figures 1,2,4,5, 6, 7, 8 and 9. Specifically, we have stated the use of Duncan's multiple range test and clarified that mean values labeled with different letters are statistically significant ($p < 0.05$). These details have been added to the respective figure legends to ensure clarity (please see lines 992-1102).

6. Reference and Citation Consistency (Please review and reconcile all citations to ensure consistency).

- Several references are cited in-text but missing from the reference list, including: Ofek et al. (2012), Kuffner et al. (2010), Grönemeyer et al. (2011), etc.
- Conversely, some references are listed but not cited (e.g., Weller 1988, Bérdy 2005).

Response: We sincerely thank the reviewer for their meticulous attention to detail in identifying the inconsistencies in our reference list. We have conducted a thorough, line-by-line review of all in-text citations and the reference list to ensure full consistency. The specific actions taken are as follows:

(1) Added missing references: The references that were cited in the text but missing from the list (Ofek et al., 2012; Kuffner et al., 2010; Grönemeyer et al., 2011) have now been added to the reference list with their full bibliographic details (please see lines 863-870, 875-878).

(2) Reconciled uncited entries:

1) The reference Weller (1988) is now correctly cited in the text (on line 71, corresponding to reference number 7 in the list).

2) The uncited reference Bérdy (2005) has been removed from the reference list.

Figures and Tables

1. Add or clarify scale bars and magnifications in SEM/TEM images (Fig. 2C-E).

Response: We thank the reviewer for this suggestion. We have verified that clear scale bars and their corresponding magnification values are already present directly beneath each individual image in Figure 2C, 2D, and 2E, forming an integral part of the figure.

2. Standardize color coding and labeling across all graphs (T1, T2, T3).

Response: We agree with the reviewer that consistent color coding is crucial for clarity. In response to this comment, we have implemented a unified color scheme for treatments T1, T2, and T3 across all relevant figures in the manuscript (Figures 4, 5, 6, 7, 8, and 9). The same specific colors are now used to represent each treatment in every graph, and the legends have been standardized accordingly. We believe this significantly improves the coherence and professional presentation of our results.

3. Ensure all axes and units are clearly labeled (e.g., g, cm, %, mg/mL).

Response: We thank the reviewer for this valuable comment. We have carefully reviewed all figures in the manuscript and have now clearly labeled all axes with their respective units as suggested (Fig.4,5 and 8).

4. Legends should be self-contained-include treatment details and significance explanations.

Response: We sincerely thank the reviewer for this critical comment. We have thoroughly revised all figure legends throughout the manuscript to ensure they are now entirely self-contained. Specifically, we have included complete treatment details and added explanations of statistical significance in all figure legends. These revisions allow readers to understand the experimental design, results and their significance directly from the figure legends (please see lines 992-1102).

Language and Grammar

1. Minor grammatical corrections are needed throughout (e.g., "founded" → "found"; "significantly reduced the abundances" → "significantly reduced the abundance").

Response: We thank the reviewer for this helpful feedback. We have performed a thorough grammatical check and revised the manuscript accordingly. The specific term "founded" mentioned by the reviewer is no longer present in the revised text. Furthermore, we have standardized the use of "abundance" (singular for the collective dataset) and "abundances" (plural for values of multiple distinct genera) throughout the manuscript to ensure consistency and accuracy.

2. Consider revising long and dense sentences for better readability.

Response: We thank the reviewer for the suggestion to improve readability. We have carefully reviewed the manuscript and revised long, dense sentences by breaking them into shorter, clearer statements. These changes have been made throughout the manuscript to enhance clarity and flow.

Discussion

1. Expand on potential applications of strain SCA4-21 in commercial biocontrol products.

Response: We thank the reviewer for this valuable input. The following sentence has been added to the Conclusion to address the commercial potential (please see Lines 464-467).

2. Briefly discuss limitations or next steps, such as testing in field conditions or formulation development.

Response: We thank the reviewer for this suggestion. The following statement has been added to the Discussion section to acknowledge this limitation and outline future work (please see Lines 462-464).

Formatting and Structure

- Reorganize long paragraphs into subheadings within the "Discussion" for better flow (e.g., "Volatile Metabolites", "Microbial Community Changes").

Response: We thank the reviewer for this excellent suggestion to improve the readability and flow of the Discussion section. We have now reorganized it with the following subheadings to provide a clear and logical structure:

(1) Antifungal activity and volatile metabolites of strain SCA4-21

(2) Biocontrol efficacy and plant growth promotion in pot experiments

(3) Reshaping of the rhizosphere bacterial community

(4) Functional reprogramming of the bacterial microbiota

(5) Modulation of the fungal community

These subheadings effectively guide the reader through the key findings, from direct antifungal mechanisms to the indirect modulation of both bacterial and fungal communities, significantly enhancing the overall clarity and narrative flow (please see Lines 334, 363, 372, 409, and 422).

Re: Spectrum02968-24R1 (Biocontrol efficiency and mechanism of novel *Streptomyces luomodensis* SCA4-21 against banana Fusarium wilt)

Dear Dr. Qiao Liu:

All comments have been addressed and I am please to recommend your manuscript for publication in Microbiology Spectrum. Please be aware that ASM policy is that data should be made available. Please upload your amplicon data to the SRA of the NCBI, and amend the corresponding data availability statement.

Your manuscript has been accepted, and I am forwarding it to the ASM production staff for publication. Your paper will first be checked to make sure all elements meet the technical requirements. ASM staff will contact you if anything needs to be revised before copyediting and production can begin. Otherwise, you will be notified when your proofs are ready to be viewed.

Sincerely,
Frédérique Reverchon
Editor
Microbiology Spectrum